

# Vegetation and geochemical responses to Holocene rapid climate change in Sierra Nevada (SE Iberia): The Laguna Hondera record

Jose Manuel Mesa-Fernández[1, 2], Gonzalo Jiménez-Moreno[1], Marta Rodrigo-Gámiz[2], Antonio García-Alix[1,2], Francisco J. Jiménez-Espejo[3], Francisca Martínez-Ruiz[2], R. Scott Anderson[4], Jon Camuera[1] and María J. Ramos-Román[1]

[1] Departamento de Estratigrafía y Paleontología, Universidad de Granada (UGR), Avda. Fuente Nueva s/n, 18002, Granada, Spain
[2] Instituto Andaluz de Ciencias de la Tierra (IACT), CSIC-UGR, Avenida de las Palmeras 4, 18100, Armilla, Granada, Spain
[3] Department of Biogeochemistry (JAMSTEC), Yokosuka, Japan.
[4] School of Earth Sciences and Environmental Sustainability, Northern Arizona University, Flagstaff, AZ, USA.

*Correspondance to:* Jose Manuel Mesa-Fernández (jmesa@iact.ugr-csic.com)

**Abstract.**

High-altitude peat bogs and lacustrine records are very sensitive to climate changes and atmospheric pollution. Recent studies show a close relationship between regional climate aridity and enhanced eolian input to lake sediments. However, changes in regional-scale dust fluxes due to climate variability at short-scales and how alpine environments were impacted by climatic- and human-induced environmental changes are not completely understood.

Here we present a multi-proxy lake sediment record of climate variability in the Sierra Nevada (SE Iberian Peninsula) over the Holocene. Palynological, geochemical and magnetic susceptibility (MS) proxies obtained from the high mountain lake record of Laguna Hondera (LH) evidence humid conditions during the Early Holocene, while a trend towards more arid conditions is recognized since ~7000 cal yr BP, with enhanced Saharan eolian dust deposition until Present. This trend towards enhanced arid conditions was modulated by millennial-scale climate variability. Relative humid conditions occurred during the Iberian Roman Humid Period (2600-1450 cal yr BP) and predominantly arid conditions occurred during the Dark Ages and the Medieval Climate Anomaly (1450-650 cal yr BP). The Little Ice Age (650-150 cal yr BP) is characterized in the LH record by an increase in runoff and a minimum in eolian input. In addition, human impact in the area is noticed through the record of *Olea* cultivation, *Pinus* reforestation and Pb pollution during the Industrial Period (150 cal yr BP-present). Furthermore, a unique feature preserved at LH is the correlation between Zr and Ca, two important elements of Saharan dust source in Sierra Nevada lake records. This supports that present day biochemical observations, pointing to eolian input as main inorganic nutrient source for oligotrophic mountain lakes, are comparable to the past record of eolian supply to these high-altitude lakes.

## 1. Introduction

Southern Spain has been the location for a number of recent studies detailing past vegetation and former climate of the region (Carrión et al., 2001, 2003, 2007, 2010; Carrión, 2002; Combourieu Nebout et al., 2009; Jiménez-Espejo et al., 2008; Martín-Puertas et al., 2008, 2010; Fletcher et al., 2010; Nieto-Moreno



et al., 2011, 2015; Rodrigo-Gámiz et al., 2011; Moreno et al., 2012 Jiménez-Moreno et al., 2015). These
studies have documented that the western Mediterranean area has been very sensitive to short-term
climatic fluctuations throughout the Holocene (e.g., Fletcher and Sánchez-Goñi, 2008; Combourieu
Nebout et al., 2009; Fletcher et al., 2010; Jiménez-Moreno et al., 2013). However, a subset of recent
studies have attempted to determine how Mediterranean alpine environments have been affected by
Holocene climate change through the study of sedimentary records from high elevation wetlands in the
Sierra Nevada (Anderson et al., 2011; García-Alix et al., 2012, 2013; Jiménez-Moreno and Anderson,
2012; Jiménez-Moreno et al., 2013; Jiménez-Espejo et al., 2014; Ramos-Román et al., 2016; García-Alix
et al., 2017). These alpine lake and bog records show minimal anthropic influence because they are
usually elevational higher than major regional Late Holocene human landscape modification. This allows
for a potentially clearer climatic signal to be determined from these sites. However, even though human
impact is less important at high-elevations, the impacts of human activities has been reconstructed from
these Late Holocene sedimentary records (Anderson et al., 2011; García-Alix et al., 2012, 2013; 2017).
Recent studies have highlighted the role of atmospheric mineral dust deposition in marine (Pulido-Villena
et al., 2008a) and terrestrial (Morales-Baquero et al., 1999; Ballantyne et al., 2011) ecosystem fertilization
through major micronutrients supply. Similar results have been described in the Sierra Nevada alpine
lakes, where Saharan dust is especially important in conditioning plankton communities from oligotrophic
lakes (Morales-Baquero et al., 2006a, 2006b; Mladenov et al., 2008; Pulido-Villena et al., 2008b; Reche
et al., 2009). Although this eolian signal has been occasionally recorded in the sedimentary sequences
from the Sierra Nevada lakes (Jimenez-Espejo et al., 2014; García-Alix et al., 2017), the record of
inorganic nutrients in Saharan dust input in past lake geochemistry has remained elusive. This study
investigates a multiproxy sediment core record from Laguna Hondera (LH), located in the Sierra Nevada
range with three main goals: (1) identifying and characterizing climatic variability during the Holocene,
focusing on vegetation changes, eolian input and runoff sediments variations; (2) understanding the
Saharan dust influence in past lake sedimentation and geochemistry, and (3) investigating the
anthropogenic impact in the area.

## 2. Study Area

Sierra Nevada is the highest mountain ranges in the southern Iberian Peninsula. Bedrock of the high
elevations of the Sierra Nevada is mostly composed of metamorphic rocks, principally mica schists
(Castillo Martín, 2009). During the late Pleistocene, the Sierra Nevada was one of the southernmost
mountains to support alpine glaciers and its last advance was recorded during the Little Ice Age (LIA;
Palma et al., 2017; Oliva et al., 2018). Subsequently to the melting of ice at the end of the Last Glacial
Maximum, wetlands and small lakes formed in the glacial cirque basins, which occur between 2451-3227
masl (Schulte, 2002; Castillo Martín, 2009; Palma et al., 2017). Several alpine wetland and lakes have
been studied in this area during the last few years as shown in Figure 1.

### 2.1. Regional Climate and Vegetation

Mediterranean climate characterises southern Iberia, with a marked seasonal variation between warm and
dry summers and cool and humid winters (e.g. Lionello et al., 2006). Overprinting this general climate is



the influence of the North Atlantic Oscillation (NAO) (Trigo et al., 2004; Trouet et al., 2009). Southern
Iberia is also characterized by strong altitudinal contrasts, which in turn controls the precipitation
patterns, with mean annual values ranging from <400mm yr -1 to >1400 mm yr -1 in the southeast desert
lowlands and the southwest highland, respectively (Jiménez-Moreno et al., 2013 and references therein),
demonstrating the complexity of climate regime in this area.
As with most mountainous regions, species and species groupings in the Sierra Nevada are distributed
with respect to elevation, depending on the temperature and rainfall gradients (e.g., El Aallali et al., 1998;
Valle, 2003). Above 2800 m the crioromediterranean flora occurs as tundra-like open grassland. The
oromediterranean belt (1900 -2800 m), mostly includes dwarf *Juniperus* (juniper), xerophytic shrublands
and pasturelands and *Pinus sylvestris* and *P. nigra*. The supramediterranean belt (~1400 - 1900 m) is
characterized by mixed deciduous and evergreen forest species (i.e., evergreen and deciduous *Quercus*,
with *Pinus spp.* and others). Mesomediterranean vegetation (600 – 1400 m), includes sclerophyllous
shrublands and evergreen *Quercus* woodlands. The natural vegetation has been strongly altered by human
activities and cultivation in the last centuries, increasing significantly the abundance of *Olea* (olive), due
to cultivation at lower altitudes (Anderson et al., 2011, and references therein), and *Pinus* due to
reforestation primarily at higher elevations (Valbuena-Carabaña, 2010).
**2.2. Laguna Hondera**
Laguna Hondera (hereafter LH; 2899 masl; 37º02.88'N, 3º17.66'W, Fig. 1) is a small and shallow lake
located at the lowest elevation of a set of lakes locally named Cañada de Siete Lagunas, a glacial valley
between two of the highest peaks of the mountain range in the Iberian Peninsula: Alcazaba (3366m) and
Mulhacén (3479m). LH has a large catchment area (154.6 ha) compared with previously studied Sierra
Nevada wetlands (Laguna de Río Seco, LdRS, 9.9 ha; Borreguil de la Caldera, BdlC, 62 ha; Morales-
Baquero et al., 1999; Ramos-Román et al., 2016). The lake was reduced to a little pond in the deepest
area of the basin when cored in September 2012, with a maximum depth of only a few centimetres.
LH presently occurs in the crioromediterranean vegetation belt (2800 m) (El Aallali et al., 1998; Valle et
al., 2003). The bedrock in the LH basin consists in Paleozoic and Precambrian mica schist with disthene
and staurolite of the lower part of the Caldera Formation (Díaz de Federico et al., 1980).
**3. Methods**
**3.1. Core sampling, lithology and chronology**
Six sediment cores were recovered from LH with a Livingstone piston corer in September 2012. LH 12-
03 (83cm) was selected for a multiproxy study because it was the longest core. Cores were wrapped with
tin foil and plastic film and transported to Universidad de Granada, where they were stored at 4ºC.
Core LH 12-03 was split longitudinally and the sediment features described. The magnetic susceptibility
was measured every 0.5 cm with a Bartington MS2E meter in SI units (x 10-4) along the entire LH 12-03
core (Fig. 2). The sediment cores were subsampled every 1 cm for different analyses, i.e., one portion for
pollen and another for geochemical analysis.



The age model was built using seven AMS radiocarbon dates from vegetal remains (Table 1; Fig. 2) by
means of Clam software (Blaauw, 2010; version 2.2), which used the IntCal13 curve for radiocarbon age
calibration (Reimer et al., 2013). The smooth spline approach was chosen (Fig. 2).  The sediment
accumulation rate (SAR) was calculated with the average rate from the Clam smooth spline output (Fig.

118    2).

**3.2. Pollen**
Pollen analysis was performed on 1 cm$^3$ of sample collected at regular 1cm interval throughout the first
62 cm of the core. Older sediments (from 62 to 82 cm depth) were barren in pollen, and only one interval
at 73 cm could be studied (Fig. 2). Pollen extraction included HCl and HF treatment, sieving, and the
addition of Lycopodium spores for calculation of pollen concentration (modified from Faegri and Iversen,
1989). Sieving was done using a 10-μm nylon sieve. The resulting pollen residue was suspended in
glycerine and mounted on microscope slides.  Slides were analysed at 400x magnification counting a
minimum of 300 pollen grains, not including the local aquatic species Cyperaceae, Ranunculaceae and
Typha. An overview of pollen taxa with abundances >1% for core LH 12-03 is plotted using Tilia
program (Grimm, 1993) in Figure 3. The pollen zonation was delimitated visually by a cluster analysis of
taxa abundance >1% using CONISS (Grimm, 1987) (Fig. 3).
**3.3. Geochemical analyses**
An X-Ray fluorescence (XRF) Avaatech core scanner®, located at the University of Barcelona, was used
to measure light and heavy elements in the LH 12-03 core. An X-ray current of 650 μA, a 10 second
count time and 10 kV X-ray voltage was used for measuring light elements, whereas 1700 μA X-ray
current, 35 second count time and 30 kV X-ray voltage was used for heavy elements.  Sampling interval
for these analyses was every 0.5 cm. For our study only three elements (K, Ca and Ti) have been
considered with enough counts to be representative.
Chemical composition was also determined on discrete samples every 2 cm. Prior to analysis, the samples
were dried in an oven and digested with HNO$_3$ and HF. Inductively coupled plasma-optical emission
spectrometry (ICP-OES; Perkin-Elmer optima 8300) was used for major element analysis. Blanks and
international standards were used for quality control – the analytical accuracy was higher than ± 2.79%
and 1.89% for 50 ppm elemental concentrations of Al and Ca, respectively, and better than ± 0.44% for 5
ppm elemental concentrations of K.
Trace element analysis was performed with an inductively coupled plasma mass spectrometry (ICP-MS;
Perkin Elmer Sciex Elan 5000). Samples were measured in triplicate through spectrometry using Re and
Rh as internal standards. The instrumental error is 2% for elemental concentrations of 50 ppm (Bea,
1996). All analyses were performed at the Instrumentations Center for Scientific Research (CIC),
University of Granada, Spain.
**3.4. Mineralogical analyses**
Morphological and compositional analyses were performed using scanning electron microscopy (SEM)
with an AURIGA model microscope (Carl Zeiss SMT) coupled with energy-dispersive X-ray





microanalysis (EDX) and Electron Backscatter Diffraction (EBSD) mode, also at the CIC (University of
Granada, Spain). Mineral grains were analysed to determine provenance, in particular those from eolian
origin.

**3.5 Statistical Analysis**

Principal components analysis (PCA) was run on the geochemical dataset using the PAST software
(Hammer et al., 2001). PCA finds hypothetical variables (components) accounting for as much as
possible of the variance in multivariate data (Davis, 1986; Harper, 1999). The elements used in the PCA
were standardized by subtracting the mean and dividing by the standard deviation (Davis, 1986). Pb was
not included in the PCA analysis due to its anthropogenic origin from mining and industrial pollution
during the latest Holocene in this area (García-Alix et al., 2013).

**4. Results**

**4.1. Lithology and magnetic susceptibility**

The LH 12-03 sediment core consists primarily of peat in the upper ~60 cm, with mostly sand and clay
layers below (Fig. 2). Positive MS peaks coincide with the grey clay intervals between 58-72 cm. Peat
intervals coincide with relatively low MS values.  For example, a minimum in MS occurs at 36-48 cm
depth, related with a peaty interval with root remains. Near the bottom of the core, between 76-80 cm, a
sandy oxidized interval occurs.

**4.2. Chronology and sedimentation rate**

The age –model of LH 12-03 documents that the record spans the last 10800 cal yr BP (Table 1; Fig. 2).
Sediment accumulation rates (SAR) were calculated using the average rate from the Clam smooth spline
output (Fig. 2). The SAR below ~39 cm is very constant, varying between 0.049 and 0.061 mm yr $^{-1}$. The
SAR increases exponentially to 0.098 mm yr $^{-1}$ at 22 cm, 0.167 mm yr $^{-1}$ at ~9 cm and 0.357 mm yr $^{-1}$ at
the core top. Accordingly with the model age and the SAR, resolution of pollen analysis varies between
~40 years per sample in the top of the core and ~120 years per sample in the lower part. The resolution of
the geochemical analysis on discrete samples changes between 100-400 years per sample, but the
geochemical XRF core scanning resolution ranges between 15-100 years per sample, providing higher
resolution than geochemical data on discrete sample. The MS analyses resolution variates between 15-
100 years per sample.

**4.3. Pollen**

Fifty distinct pollen taxa were recognized, but only those with abundance higher than 1% are included in
the pollen diagram (Fig. 3). Five pollen zones for the LH 12-03 record are identified, using variation in
pollen species plotted in Figure 3 and a cluster analysis run through the program CONISS (Grimm, 1987).
Zone LH-1 (core bottom-7000 cal yr BP) is defined by only three samples, due to the low preservation of
pollen in this interval.  Pollen in this zone is dominated by an alternation between Asteraceae and *Pinus*
(Fig. 3).  Arboreal pollen (AP), composed primarily of *Pinus*, but also *Quercus*, reaches its maximum





occurrence (90%) at ~7000 cal yr BP. The highest occurrence of Onagraceae pollen (~10%) takes place in
this zone, and Caryophyllaceae reaches high values during this zone (~10%) as well. Only minor
amounts of graminoids (Poaceae and Cyperaceae) occur during this period.
Zone LH-2 (~7000-4000 cal yr BP) is characterized by high percentages of tree species, primarily *Pinus*,
at the beginning of the zone (~90%), decreasing to ~55% at the upper part of the zone, with a minimum
(~30%) at 5000 cal yr BP. Quercus increases from ~2% at the beginning of the zone to ~10% at the end.
The highest percentages of *Betula* pollen (~5%) in the record occurs at this time. Asteraceae pollen (~5-
30%) is less than in LH-1, but Poaceae increases from <5% at the opening of the zone to >25%.
Caryophyllaceae and Onagraceae continue to show relatively high values in this zone (~5% and ~6%,
respectively). Cyperaceae occurs in high percentages (15%).
Zone LH-3 (~ 4000-2600 cal yr BP) is defined primarily by a great increase in Poaceae pollen (to ~60%)
(Fig. 3). Other important herbs and shrubs include Asteraceae (5-15%) and Caryophyllaceae (~5%).
Other pollen types that increase for the first time in this zone include Ericaceae (~3%), *Artemisia* (~3%)
and Ranunculaceae (~2-6%). *Pinus* (~3-25%) and Cyperaceae (0-14%) record a minimum in this zone,
and Onagraceae disappears altogether (Fig. 3).
Zone LH-4 (~ 2600-1450 cal yr BP) pollen assemblages show high variability in this zone. *Pinus* variates
between ~80% to ~3% from the onset to the end of the zone. Aquatic pollen such as Cyperaceae (~15%)
increases. On the other hand, an increase in herbs as Asteraceae (~5-70%) occurs along the zone, Poaceae
variates between ~7-12%.
Zone LH-5 (~ 1450-600 cal yr BP) is characterized by an increase in herbaceous pollen, led by Poaceae
(~35% maximum during this zone), Asteraceae (~60% maximum during this zone after ~1000 cal yr BP)
and *Artemisia* (~10%), and with the resulting decrease in AP. Since this zone to the present *Quercus* is
the major component of AP instead of *Pinus*. Cyperaceae also shows a decrease, and Ranunculaceae
reaches ~ 5%.
Zone LH-6 (~ 600 cal yr BP-present) is divided in two subzones. LH-6A (~ 600- 150 cal yr BP)
documents an increase in *Olea* (~6%), Poaceae (20%), Caryophyllaceae (7%) and *Artemisia* (~2-20%).
*Pinus* (~2%) and Asteraceae (~60%) decrease in this period. Aquatic and wetland pollen show a rise
(Cyperaceae ~30%, Rannunculaceae ~10%). LH-6B (~ 150 cal yr BP-present) depicts a further increase
in *Olea* (~25%), Poaceae (~40%) and *Artemisia* (~10%).
**4.4. Sediment composition**
Results of the geochemistry are described following the pollen zonation previously defined (see above).
The XRF-scanning method relies on determining the relative variations in elements. Nevertheless the
presence of major variations in organic matter or carbonates makes it important to normalize the
measured count in order to obtain an environmentally relevant signal (Löwemark et al., 2011).
Aluminium and titanium normalizations are commonly used to discern possible fluctuations in the
lithogenic fraction (enrichment or depletion of specific elements), particularly in the terrigenous
aluminosilicate sediment fraction (Van der Weijden, 2002; Calvert and Pedersen 2007; Martinez-Ruiz et
al., 2015). For this study, the XRF data were normalized to Ti since Al counts obtained were very low.
Poor detection of Al can be related to either low Al content, or high organic and water content that



increase radiation absorption and affect the intensity of this light element, among other possibilities
(Tjallingii et al., 2007).
Since data spacing is different between the analyses on discrete samples and the XRF scanner, a linear
interpolation was performed with the purpose of equalizing the space of the different time series (150-300
years). Afterwards, the mobile average was worked out along the time series (taking into account the 5
nearest points) in order to easily identify trends by means of smoothing out data irregularities. The
obtained data were compared, and both XRF-scanner and discrete sample data showed a good correlation.
As a consequence, the geochemical proxies displayed higher time resolution than the discrete samples
(Table 2).  Discrete sample and XRF data results are described together in order to simplify this section
(Fig. 4).
Zone LH-1 (core bottom- ~7000 cal yr BP) is typified by maximum values of K/Al and K/Ti ratios,
coinciding with the lowest values in Ca/Al, Ca/Ti and Zr/Al ratios. Pb/Al data show a stable pattern
during this interval. Nevertheless, between 10000-9000 cal yr BP and ~8200 cal yr BP the trends were
reversed, with relatively low K/Al, low K/Ti and slightly increasing Zr/Al, Ca/Al and Ca/Ti ratios. A
positive peak in Pb/Al ratio at ~8200 cal yr BP is also observed.
Zone LH-2 (~7000-4000 cal yr BP) shows a decreasing trend in K/Al and K/Ti ratios, while an increasing
trend in Zr/Al, Ca/Al and Ca/Ti occurred. The Pb/Al ratio remains constant throughout this zone.
Zone LH-3 (~4000-2600 cal yr BP) documents an increase in Zr/Al, Ca/Al and Ca/Ti ratios, which
reaches a maximum at ~2600 cal yr BP. A K/Al and K/Ti minima occurs between ~3000 and ~2600 cal
yr BP. The Pb/Al ratio shows a positive peak at ~2800 cal yr BP.
Zone LH-4 (~2600-1450 cal yr BP) is characterized by low Ca/Al, Ca/Ti and Zr/Al ratios, with relatively
high K/Al and K/Ti ratios. The Pb/Al ratio shows a flat pattern, increasing at ~1500 cal yr BP.
Zone LH-5 (~1450- 650 cal yr BP) depicts higher ratios of Zr/Al, Ca/Al and Ca/Ti and decreasing ratios
of K/Al and K/Ti. A somewhat higher Pb/Al ratio is also registered during this period.
Zone LH-G6 (~ 650 cal yr BP- present) is divided in two subzones. During the LH-G6a subzone, low
values of Zr/Al and Ca/Ti ratios and minimum values Ca/Al ratio occur. Higher K/Al and K/Ti values are
also observed. The Pb/Al ratio decreases during this interval. LH-G6b is characterized by Zr/Al, Ca/Al,
Ca/Ti, K/Ti and Pb/Al maxima. Lower K/Al ratio occurs in this zone.
Several studies have demonstrated that PCA analysis of geochemical data can elucidate the importance of
different geochemical components driving the environmental responses in marine and lacustrine records
(Bahr et al., 2014; Yuan, 2017).  We performed a PCA analysis of the LH geochemical data, which
yielded two significant components (Fig. 5).  The first principal component (PC1) describes 58% of the
total variance. The main negative loadings for PC1 are Rb, Ba, Al, K, Ca, Mg and Sr, while large positive
loadings correspond to Zr and Rare Earth Elements (REE).  The second principal component (PC2)
explains 17% of the total variance. The main negative loading for PC2 are Fe, Ca, Zr, Mg and Lu.
Positive loads correspond to Al, K, Ba, Sr and other elements.
SEM analyses show an alternation between a lithology rich in rock fragments and another rich in organic
remains. Also, diatom frustules, rich in silica, are particularly abundant since ~6300 cal yr BP to Present.
Other minerals such as zircon, rounded quartz and monazite were also identified (Fig.6).



**5. Discussion**
Pollen and geochemical proxies have been widely used for reconstructing vegetation changes and
environmental and climate variations in southern Iberia (e.g. Carrión, 2002; Sánchez-Goñi and Fletcher,
2008; Anderson et al., 2011; Nieto-Moreno et al., 2011; Jiménez-Moreno et al., 2012; Moreno et al.,
2012; Fletcher and Zielhofer, 2013; Jiménez-Espejo et al., 2014; Ramos-Román et al., 2016). Variations
in the occurrences of arboreal taxa such as *Pinus* and other mesic species (e,g, *Betula*, *Quercus*),
indicating relative humid and warm conditions, and xerophytic species (e.g., Poaceae, Asteraceae,
Amaranthaceae, *Artemisia*), representing aridity, have been useful for reconstructing relative humidity
changes in southern Iberian (e.g. Carrión et al., 2001, 2007, 2010; Anderson et al., 2011; Jiménez-Moreno
et al., 2012, 2013, 2015; Ramos-Román et al., 2016).
Over 75% of the total geochemical data variance is explained by the PC1 and PC2 (Fig. 5). We interpret
the results of PC1 as resulting from certain sorting between heavy minerals (positive loading; Zr and
REE) vs. clay minerals and feldspars (negative loadings; K, Al and Ca). The drainage basin is composed
mainly by mica schist, consequently enhanced in K-rich minerals such as mica and feldspar (Díaz de
Federico et al., 1980). PC1 points to a sorting between heavy minerals (enriched in Zr and REE) and clays
and feldspars (enriched in K and Al) (Fig. 5a), probably linked to physical weathering within the basin
and to resulting runoff until final deposition in the lake.
On the other hand, we interpret the results of PC2 as differentiating autochthonous elements (positive
loadings) vs. Saharan allochthonous input (negative loadings). In the first case, due to the abundance of
mica schist within the LH drainage basin (Díaz de Federico et al., 1980), the K/Al and K/Ti ratios are
interpreted as detrital products, and thus a proxy of runoff. In the second case, PC2 negative loading Zr,
Ca, Mg and Fe (Fig. 5b) grouped elements that are coherent with Saharan input composition (dolomite,
iron oxides and heavy minerals) (Ávila, 1997; Morales-Baquero et al., 2006b; Pulido-Villena et al.,
2007). In addition, Ca shows a strong positive correlation with Zr since 6300 cal yr BP (r =0.57; p<0.05)
supporting an eolian origin of the Ca in LH sediments. For instance, enrichment in heavy minerals such
as zircon and palygorskite has previously been used as an eolian proxy in the western Mediterranean (e.g.,
Combourieu Nebout et al., 2002, Rodrigo-Gámiz et al., 2011, 2015). High concentrations of Ca in other
lacustrine systems is usually associated with biogenic sources when anti-correlated with terrigenous
elements (Yuan, 2017). Nevertheless, elevated Ca in the LH record is linked with detrital elements, as
shown by PC1, where Ca is associated with K and Al (Fig. 5a). For these reasons Ca/Al and Ca/Ti ratios
are used in LH as eolian input proxies.
Elemental ratio variations, such as the ratios K/Al and K/Ti indicating fluvial input and ratios Zr/Al or
Zr/Th indicating aridity and eolian input, have been previously interpreted from Alboran Sea records as
well as in southern Iberia (Martín-Puertas et al., 2010; Nieto-Moreno et al., 2011, 2015; Rodrigo-Gámiz
et al., 2011; Jiménez-Espejo et al., 2014; Martínez-Ruiz et al., 2015; García-Alix et al., 2017). Thus, the
integration of both palynological data and geochemical ratios used as detrital input from LH have allowed
the reconstruction of the palaeoclimate and palaeoenvironmental history in Sierra Nevada during the
Holocene.
**5.1. Holocene palaeoclimate and palaeoenvironmental history**



### 5.1.1. Early and Mid-Holocene humid conditions (core bottom – ~7000 cal yr BP)

The wettest conditions are recorded during the Early Holocene in Sierra Nevada. This is shown in the LH record by the highest K/Al ratio and MS values, and the low values in Zr/Al, Ca/Al and Ca/Ti ratios, suggesting that runoff dominated over eolian processes at this time (zone LH-1; Fig. 7) and agrees with previous studies in the area (Anderson et al., 2011; Jiménez-Moreno and Anderson, 2012; García-Alix et al., 2012; Jiménez-Espejo et al., 2014). Unfortunately, the pollen record from LH during this interval is insufficient to definitely confirm this interpretation, due to the high detrital sediment composition and low organic content, as shown by the low MS values and low pollen preservation. However, high percentages of AP in two out of three analysed samples suggest humid conditions and high runoff during this period.

An Early Holocene humid stage is noticed in other nearly sites, such as the south-faced Laguna de Río Seco (LdRS; Fig. 1) (Anderson et al., 2011), when the highest lake level of the Holocene occurred. This is also coeval with the dominance of arboreal species such as *Pinus* as well as aquatic and wetland plants (Anderson et al., 2011). Low eolian input, noted by geochemical ratios, is also recorded in LdRS during this interval (Jiménez-Espejo et al., 2014). Further indications of elevated humidity come from the north-facing Borreguil de la Virgen (BdlV) (see Fig. 1), which is dominated by an AP assemblage and a high occurrence of aquatic algae *Pediastrum* along with a higher lake level (Jiménez-Moreno et al., 2012).

Although the preponderance of evidence accumulated for the Early Holocene suggests overall humid conditions, at least three relatively arid periods are identified with the geochemical data in the LH record (Fig. 7). The first arid period occurred between ~9600-9000 cal yr BP, the second occurred ~8200 cal yr BP and the third around 7500 cal yr BP.

The first arid event is characterized in LH by a decrease in K/Al and K/Ti ratios and MS, resulting from the lower runoff input with the concomitant change to a more peaty composition. This event could be correlated with a dryness event recorded in the Siles Lake record (Carrion, 2002) at ~9300 cal yr BP noticed by an increase in *Pseudoschizaea*, which was coeval with a minor decrease in arboreal pollen also recorded in several sites in North Iberia (Iriarte-Chiapusso et al., 2016). At marine site ODP 976 (Fig.1; Combourieu-Nebout et al., 2009) a decrease in deciduous *Quercus* occurred between 9500-9200 cal yr BP indicating a rapid excursion towards arid conditions (Fig.7). The speleothem record of Corchia Cave also shows dryer conditions during this interval (Fig. 7; Regattieri et al., 2014) In addition, a decrease in fluvial input in the Southern Alps and an aridification phase in southeastern France and southeastern Iberia has been similarly recorded (Jalut et al., 2000).

The second dry event recorded at ~8200 cal yr BP is depicted in LH record by a negative peak in K/Ti and K/Al ratios, and by the onset of a trend toward peatier lithology as evidenced by the MS profile. This event is not recognized in LH record as clearly as the 9500 cal yr BP and the 7500 cal yr BP dry events. A decrease in *Pinus* percentage is observed in the nearby LdRS (Anderson et al., 2011), while a forest decrease is recorded in the Alboran Sea sites MD95-2043 and ODP 976. In several records from north western Iberia a decrease in arboreal pollen also occurred at this time (Iriarte-Chiapusso et al., 2016).

The 8.2 ka event was the most rapid climate change towards cooler conditions occurred during the Holocene. It was defined in Greenland ice cores by minimum values in δ18O and affected the North Atlantic basin and the Mediterranean area (Alley et al., 1997; Rasmussen et al., 2007; Wiersma et al., 2011). Recent simulations point to a fresh water input in North Atlantic which could slow down the North



Atlantic Deep Water (NADW) formation preventing the heat transport over the north hemisphere
(Wiersma et al., 2010, 2011; Young et al., 2013).
Another dry event is recorded in LH at ~7500 cal yr BP evidenced by the higher peat content in the
sediment, as well as by the lower MS values and a relative minimum in the K/Ti ratio. A relative AP
minimum also occurred in LH at this time. This short-live event are depicted sharper than 8200 cal yr BP
event in several sites in southern Iberia and Alboran Sea: In the Padul record, located at 744 masl at the
lower part of Sierra Nevada a decrease in both evergreen and deciduous *Quercus* is interpreted as a dry
and cold event (Ramos-Román et al., in review); forest expansion in Guadiana valley during the early-
mid Holocene is interrupted by a xeric shrublands development between 7850-7390 cal yr BP (Fletcher et
al., 2007); in the Alboran Sea a decrease in deciduous *Quercus* is registered at site MD95-2043; at site
300G a decrease in winter and summer temperatures is also recorded during this interval (Jiménez-Espejo
et al., 2008); in lake Pergusa (south Italy) a trend toward arid conditions began at ~7500 cal yr BP
(Magny et al., 2012); in Corchia Cave an arid excursion occurred at ~7500 cal yr BP within an overall
humid period between 8300 cal yr BP and 7200 cal yr BP (Fig. 7; Regattieri et al., 2014).
Importantly, these arid events recorded in LH at 9600-9000 cal yr BP and 8200 cal yr BP are coeval with
the ice-rafted debris events 6 and 5 defined by Bond et al. 1997 in North Atlantic.

**5.1.2. Mid- and Late Holocene (~7000 cal yr BP-2600 cal yr BP)**

The Middle and Late Holocene in the southern Iberian Peninsula is characterized by a trend towards more
arid conditions (Jalut et al., 2009; Anderson et al., 2011; Rodrigo-Gámiz et al., 2011; Jiménez-Moreno
and Anderson, 2012; Jiménez-Espejo et al., 2014). In the LH record an abrupt decrease in the MS values
indicates a lithological change to more peaty sedimentation at ~7000 cal yr BP. Similarly, a decrease in
the K/Al and K/Ti ratios, points to a transition to less humidity and runoff (Fig. 7). *Quercus* percentages
increase at this time, partially replacing the *Pinus* which mainly compose the AP during the record. A
progressive increasing trend in eolian input from Sahara (Zr/Al, Ca/Al and Ca/Ti ratios) is observed
around 5500-6500 cal yr BP (Fig. 7), also pointing to an increase in aridity in the area. This change
coincides with regional increases in the Zr/Th ratio (equivalent to Zr/Al ratio) and *Artemisia* pollen, and
with decreases in *Betula* and *Pinus* in the LdRS record (Anderson et al., 2011; Jiménez-Espejo et al.,
2014), and in *Pinus* in the BdlV record (Jiménez-Moreno et al., 2012). Rodrigo-Gámiz et al. (2011) and
Jiménez-Espejo et al. (2014) observed similar geochemical patterns in western Mediterranean marine
records and in LdRS, with a decline in fluvial input, and a decline in surface runoff, respectively. The
same pattern is noticed in marine pollen records MD95-2043 and ODP 976 (Fletcher and Sanchez-Goñi,
2008; Combourieu-Nebout et al., 2009; Fig. 7). Contemporaneously, aridity is also suggested from
speleothem data around the Mediterranean area: At El Refugio cave, a hiatus in the speleothem growing
rate occurred between 7300-6100 cal year BP (Walczak et al., 2015), which is coeval with a drop in δ18O
in Soreq (Israel) and Corchia (Italy; CC26; Fig. 1 and 7) caves at 7000 cal yr BP (Bar-Matthews et al.,
2000; Zanchetta et al., 2007; Regattieri et al., 2014). Also at ~7000 cal yr BP a decreasing trend in the
deciduous/sclerophyllous pollen ratio occurred in southeastern France and Iberia (Jalut et al., 2000) and at
continental sites around the Mediterranean Sea (Jalut et al., 2009). In addition, very low lake levels were
recorded in the Sahara-Sahel Belt (Liu et al., 2007) and in the Southern Alps (Magny et al., 2002).



Enhanced arid conditions are observed in the LH record between 4000-2500 cal yr BP, interpreted
through a decline in AP, a Poaceae maximum and a peak in *Artemisia*. Also a surface runoff minimum
and an increase in eolian input proxies took place between 3500-2500 cal yr BP (zone LH-3). In Corchia
Cave an arid interval was recorded at ~3100 cal yr BP (Regattieri et al., 2014), coeval with another one
observed globally and described by Mayewski et al. (2004) between 3500-2500 cal yr BP. Nevertheless,
this period is not climatically stable, fluctuations are observed in in K/Ti, K/Al, Ca/Ti, Ca/Al and Zr/Al
ratios. Furthermore, peaks in *Quercus* are recorded in LH, LdlM and ODP 976 sites at ~3900 cal yr BP
and ~3100 cal yr BP, when AP in LH decreases (Combourieu-Nebout et al., 2009; Jiménez-Moreno et al.,
2013). This fact a priori contradictory, could be explained by altitudinal displacements of the tree taxa
such as *Quercus* in the oromediterranean belt due to the climatic variability occurred along this interval
(Carrión, 2002). During warmer periods, this species would be displaced towards higher elevation and the
influence of *Quercus* pollen in Sierra Nevada would be larger, this could explain relative
higher *Quercus* percentages in LdlM, LH and also in the ODP 976 record. The same relationship
between *Quercus* and *Pinus* is observed comparing the BdlC and Padul records, located closely but with
large altitude difference (BdlC ~2992 masl; Padul ~725 masl; Ramos-Román, 2018) where is also likely
linked to movements in the oromediterranean belt (Ramos-Román, 2018). These altitudinal displacements
of the tree taxa have been previously related to temperature changes in others southern Iberian records,
suggesting an ecological niche competition between *Pinus* and *Quercus* species at middle altitudes (see
Carrión et al., 2002 for a revision).
**5.1.3. Iberian Roman Humid Period (IRHP; ~2600-1450 cal yr BP)**
Because there is no consensus in the literature about the chronology for the main climatic stages during
the last 2000 years (Muñoz-Sobrino et al., 2014; Helama et al., 2017), here we follow the chronology
proposed by Moreno et al. (2012): Dark Ages (DA, 1450-1050 cal yr BP); Medieval Climate Anomaly
(MCA, 1050-650 cal yr BP); and LIA (650-100 cal yr BP). Another climatic stage preceeds the DA – the
Iberian Roman Humid Period (IRHP, 2600-1600 cal yr BP), originally described by Martín-Puertas et al.
(2008). However, in the LH record we have established different IRHP limits (2600-1450), based
accordingly to the pollen zonation (Fig. 3), and coinciding with the DA onset defined by Moreno et al,
409    (2012).
The IRHP has been described as the wettest period in the western Mediterranean from proxies determined
both in marine and lacustrine records during the Late Holocene (Reed et al., 2001; Fletcher and Sanchez-
Goñi 2008; Combourieu-Nebout et al., 2009; Martín-Puertas et al., 2009; Nieto-Moreno et al., 2013;
Sánchez-López et al., 2016). A relative maximum in AP occurred in the LH record during this time, also
indicating forest development and relative high humidity during the Late Holocene in the area (zone LH-
4; Fig. 7). This is further supported by high K/Al and K/Ti ratios and MS values, indicating high detrital
input in the drainage basin, a minimum in Poaceae and low Saharan eolian input (low Ca/Al, Ca/Ti and
Zr/Al ratios) (Fig. 7). Fluvial elemental ratios have also shown an increase in river runoff in Alboran Sea
marine records (Nieto-Moreno et al., 2011; Rodrigo-Gámiz et al., 2011). This humid period seems to be
correlated with a solar maximum (Solanki et al., 2004) and persistent negative NAO conditions (Olsen et
al., 2012), which could have triggered general humid conditions in the Mediterranean. However, in the
LH record a decrease in AP between 2300-1800 occurred, pointing to arid conditions at that time. This



arid event also seems to show up in BdlC, with a decrease in AP between 2400-1900 cal yr BP (Ramos-
Román et al., 2016) and in Zoñar Lake, with water highly chemically concentrated and gypsum
deposition between 2140-1800 cal yr BP (Martín-Puertas et al., 2009). In Corchia Cave a rapid excursion
towards arid condition is recoded at ~2000 cal yr BP (Regattieri et al., 2014) (Fig.7).
**5.1.4. Dark Ages and Medieval Climate Anomaly (DA, MCA; 1450-650 cal yr BP)**
Predominantly arid conditions, depicted by high abundance of herbaceous and xerophytic species and an
AP minimum in the LH record, are shown for both DA and MCA (zone LH-5; Fig. 7). This is further
supported in this record by an increase in Saharan eolian input Ca/Al, Ca/Ti and Zr/Al ratios, and by a
decrease in surface runoff, indicated by the K/Al and K/Ti ratios (zone LH-5; Fig. 7). These results from
LH agree with climate estimations of overall aridity modulated by a persistent positive NAO phase during
this period (Trouet et al., 2009; Olsen et al., 2012), also previously noted by Ramos-Román et al. (2016)
in the area (Fig. 7).
Generally arid climate conditions during the DA and the MCA have also been previously described in the
LdlM and BdlC records, shown by a decrease in mesophytes and a rise of xerophytic vegetation during
that time (Jiménez-Moreno et al., 2013; Ramos-Román et al., 2016). Several pollen records in south and
central Iberian Peninsula also indicate aridity during the DA and MCA, for example grassland expanded
at Cañada de la Cruz, while in Siles Lake a lower occurrence of woodlands occurred (Carrión, 2002).
Also in Cimera Lake low lake level and higher occurrence of xerophytes were recorded (Sánchez-López
et al., 2016). Arid conditions were depicted in Zoñar Lake by an increase in *Pistacia* and heliophytes (i.e.,
Chenopodiaceae) and lower lake level (Martín-Puertas et al., 2010). Similar climatic conditions were
noticed in the marine records MD95-2043 and ODP 976 in the Alboran Sea through decreases in forest
(Fletcher and Sánchez-Goñi, 2008; Combourieu-Nebout et al., 2009; Fig. 7). Arid conditions in Basa de
la Mora (northern Iberian Peninsula) occurred during this time, characterized by maximum values of
*Artemisia*, and a lower development of deciduous *Quercus* and aquatic species such as *Potamogeton*, also
indicating low lake water levels (Moreno et al., 2012). Arid conditions were also documented by
geochemical data in marine records from the Alboran Sea (Nieto-Moreno et al., 2013, 2015), in the Gulf
of Lion and South of Sicily (Jalut et al., 2009). Aridity has also been interpreted for central Europe using
lake level reconstructions (Magny, 2004) and in speleothems records in central Italy (Regattieri et al.,
450 2014).

**5.1.5. Little Ice Age (LIA; 650-150 cal yr BP)**
The LIA is interpreted as an overall humid period in the LH record. This is indicated by higher AP values
than during the MCA, low Saharan dust input (low Ca/Al, Ca/Ti and Zr/Al ratios), a decrease in herbs
(Poaceae) and high values in the K/Al and K/Ti ratios indicating enhanced runoff (zone LH-6A; Fig. 7).
An increase in fluvial-derived proxies has been previously documented in other Iberian terrestrial records
such as Basa de la Mora Lake (Moreno et al., 2012), Zoñar Lake (Martín-Puertas et al., 2010) or Cimera
Lake (Sánchez-López et al., 2016) and marine records from the Alboran Sea basin (Nieto-Moreno et al.,
2011, 2015). Lake level reconstructions in Estanya Lake, in the Pre-Pyrenees (NE Spain), have shown
high water levels during this period (Morellón et al., 2009, 2011), supporting our humid climate



inferences. Nevertheless, recent high-resolution studies in Sierra Nevada (Ramos-Román et al., 2016;
García-Alix et al., 2017) and in several Iberian mountains (Oliva et al., 2018) have revealed that LIA was
not a climatically stable period and many oscillations at short-time scale occurred.
A persistently negative NAO phase, although with high variability, occurred during this time period
(Trouet et al., 2009), which could explain the overall humid conditions observed in southern Europe. As
in the Early Holocene arid events, solar variability has been hypothesized as the main forcing of this
climatic event (Bond et al., 2001; Mayewski et al., 2004; Fletcher et al., 2013; Ramos-Román et al.,
467 2016).

### 5.2. Anthropogenic impact in the southern Iberia

Previous studies, including the nearby LdRS record in Sierra Nevada, have shown that mining and
metallurgy activities commenced by ~4500 cal yr BP in this area (García-Alix et al., 2013, and references
therein), as shown by an enhanced Pb/Al ratio since this time. For the LH record, the first clear signal of
lead pollution from mining and smeltering occurred around 2800 cal yr BP, coinciding with the Late
Bronze Age (LBA) (3200-2800 cal yr BP) and the Early Iron Age (EIA) (2800-2500 cal yr BP) (zone
LH-3; Fig. 8). The same signal is also recorded in the nearby LdRS (García-Alix et al., 2013; Fig 8).
Many studies, including LdRS, have shown that the IRHP was the most important lead pollution period
prior to the IP (Settle and Patterson, 1980). However, the Pb/Al record from LH does not register
enhanced pollution at this time. This could be due to a local effect, such as a higher catchment area in LH
involving a high runoff input, supported by an increase in the K/Al and K/Ti ratios during this humid
period that could have diluted the Pb signal transported by eolian input. Also a regional effect, such as a
weaker dust mobilization due to the humid conditions prevailing at this time, or patchy pollution
distribution, could explain these diverse records.
An increasing trend in *Artemisia*, which points to a climatic or anthropic aridification, is coeval with
another Pb/Al peak that occurred during the MCA (Fig. 8). Increasing anthropic activities during this time
in the area are justified by the first appearance of coprophilous fungi such as *Sordiales* and *Sporormiella*,
which occurred in BdlC (Ramos-Román et al., 2016) and in LdRS (Anderson et al., 2011), suggesting
grazing activity at high altitudes in Sierra Nevada (Anderson et al., 2011; Jiménez-Moreno and Anderson,
2012; Ramos-Román et al., 2016). Maxima in *Artemisia* and coprophilous fungi in Sierra Nevada are also
reached during the last 500 years (Anderson et al., 2011; Jiménez-Moreno and Anderson, 2012; Ramos-
Román et al., 2016).
An increase in the Pb/Al ratio is recorded during the IP in the LH record (Fig. 8), suggesting more
mining, fossil fuel burning or other human industrial activities. This is coeval with a rise in AP, which is
also related to human activities such as *Olea* commercial cultivation at lower elevations around Sierra
Nevada or *Pinus* reforestation in the area (Valbuena-Carabaña et al., 2010; Anderson et al., 2011). The
same pattern has also been observed in others records from Sierra Nevada (Jiménez-Moreno and
Anderson, 2012; García-Alix et al., 2013; Ramos-Román et al., 2016), in Zoñar Lake and the Alboran Sea
records (Martín-Puertas et al., 2010). In addition, a progressively increasing trend in Zr/Al and Ca/Al
ratios is observed during the last two centuries, which could be related to increasing local aridity and/or
anthropogenic desertification, but also with a change in the origin and/or composition of the dust reaching



to the lake (Jiménez-Espejo et al., 2014), likely related to the beginning of extensive agriculture and the
concomitant desertification in the Sahel region (Mulitza et al., 2010).
Therefore, the human impact in LH is mostly remarkable during the last two millennia. The comparison
with nearby records such as LdRS has also revealed that high-mountain lakes are very sensitive to human
activities (Anderson et al., 2011).
**5.3 Significance of the eolian record from Laguna Hondera**
Saharan dust influence over current alpine lake ecosystems is widely known (Morales-Baquero et al.,
2006a, 2006b; Pulido-Villena et al., 2008b; Mladenov et al., 2011), nevertheless, none of the previous
record preserved the relationship between elements found in present-day Saharan dust. The most
representative elements of Saharan dust in LH record are Fe, Zr and Ca as shown by the PC2 loading
(Fig. 5), where Ca and Fe directly affect the alpine lake biogeochemistry in this region (Pulido-Villena et
al., 2006, 2008b). Zirconium is transported in heavy minerals in eolian dust (Govin et al., 2012) and has
largely been used in the Iberian Peninsula and the western Mediterranean as an indicator of eolian
Saharan input (Moreno et al., 2005; Nieto-Moreno et al., 2011; Rodrigo-Gámiz et al., 2011; Jiménez-
Espejo et al., 2014; Martínez-Ruiz et al., 2015, and references therein). High Zr content has also been
identified in present aerosols at high elevations in Sierra Nevada (García-Alix et al., 2017). Considering
the low weatherable base cation reserves in LH bedrock catchment area, calcium is suggested to be
carried by atmospheric input of Saharan dust into alpine lakes in Sierra Nevada (Pulido-Villena et al.,
2006, see discussion; Morales-Baquero et al., 2013). This is the first time that the Ca signal is properly
recorded in a long record from Sierra Nevada. This could be explained by higher evaporation rates at this
site promoting annual lake desiccation that could prevent Ca water column dissolution and
using/recycling by organism, preserving better the original eolian signal. These elements have an essential
role as nutrients becoming winnowed and recycled rapidly in the oligotrophic alpine lake ecosystem
(Morales-Baquero et al., 2006b). This phenomenon has also been observed in other high-elevation lakes
where the phytoplankton is supported by a small and continually recycled nutrient pool (e.g., Sawatzky et
al., 2006).
The SEM observations further confirm the presence of Saharan dust in the lake sediments from LH and
the occurrence of Zircon, the main source of eolian Zr, which is relatively abundant (Fig. 6a). Quartz with
rounded morphologies (eolian erosion) are also frequent (Fig. 6b) in the uppermost part of the record as
well as REE rich minerals, such as monazite, which is typical from the Saharan-Sahel Corridor area
(Moreno et al., 2006) (Fig. 6c). In addition, the fact that the highest correlation between Ca and Zr
occurred after ~6300 cal yr BP, (r=0.57 p<0.005) along with the SEM observation and the low
availability of Ca in these ecosystems, could suggest that the beginning of Saharan dust arrivals to the
lake including both elements took place at this time, giving rise to the present way of nutrient inputs in
these alpine lakes (Morales-Baquero et al., 2006b; Pulido-Villena et al., 2006). The onset of Saharan dust
input into southern Iberia occurred prior to the end of the African Humid Period (AHP; ~5500 cal yr BP;
deMenocal et al., 2000), as previously noticed in the nearby LdRS (Jiménez-Espejo et al., 2014) and in
Alboran Sea (Rodrigo-Gámiz et al., 2011). This could suggest a progressive climatic deterioration in



North Africa, which culminated with the AHP demise and the massive Saharan dust input recorded in all
records in Sierra Nevada at ~3500 cal yr BP (Fig. 7).

## 6. Conclusions


The multiproxy paleoclimate analysis from LH has allowed the reconstruction of the vegetation and
climate evolution in Sierra Nevada and southern Iberia during the Holocene, and the possible factors that
have triggered paleoenvironmental changes. Climate during the Early Holocene was predominantly
humid, with two relatively arid periods between 10000-9000 and ~ 8200 cal yr BP, resulting in less
detrital inputs and a change to more peaty lithology.  The onset of an arid trend took place around 7000
cal yr BP, decreasing the runoff input in the area. A significant increase in eolian-derived elements
occurred between 6300-5500 cal yr BP, coinciding with the AHP demise. An arid interval is recorded
between 4000-2500 cal yr BP, with a vegetation assemblage dominated by xerophytes.
Relative humid conditions occurred in the area between 2500-1450 cal yr BP, interrupting the Late
Holocene aridification trend.  This humid interval was characterized by expansion of forest vegetation,
high runoff input, and a more clayey lithology.  But during the DA and the MCA (1450-650 cal yr BP)
there was enhanced eolian input and an expansion of xerophytes, indicating increased arid conditions. In
contrast, the LIA (650-150 cal yr BP) was characterized by predominant humid conditions as pointed out
high runoff and low eolian input.
The first human impact signals in LH is recorded at ~2800 cal yr BP with a rise of Pb/Al ratio, coinciding
with the onset of mining in the Iberian Peninsula. The IP (150 cal yr BP-Present) is characterized in the
LH record by the highest values of the Pb/Al ratio, indicating fossil fuel burning by metallurgy industry,
enhanced of mining and other human activities.
Importantly, the LH record shows a unique and exceptional Ca signal derived from eolian input (high Ca-
Zr correlation) during the past ~6300 years in Sierra Nevada. The good preservation of the Ca record
might have been favoured by the high evaporation and the low lake depth that could have prevented Ca
column water dissolution and its re-use by organisms. Our record indicate that present-day inorganic
nutrient input from Sahara was established 6300 yrs ago and lasted until the present, with variations
depending on the prevailing climate.

## Acknowledgements


This study was supported by the project P11-RNM 7332 of the "Junta de Andalucía", the projects
CGL2013-47038-R, CGL2015-66830-R of the "Ministerio de Economía y Competitividad of Spain and
Fondo Europeo de Desarrollo Regional FEDER", the research groups RNM0190 and RNM179 (Junta de
Andalucía). We also thank to Unidad de Excelencia (UCE-PP2016-05). J.M.M.F acknowledge the PhD
funding provided by Ministerio de Economía y Competitividad (CGL2015-66830-R) A.G.-A. was also
supported by a Marie Curie Intra-European Fellowship of the 7th Framework Programme for Research,
Technological Development and Demonstration of the European Commission (NAOSIPUK. Grant
Number: PIEF-GA-2012-623027) and by a Ramón y Cajal Fellowship RYC-2015-18966 of the Spanish
Government (Ministerio de Economía y Competitividad) and M.R.G. from the Andalucía Talent Hub
Program co-funded by the European Union's Seventh Framework Program (COFUND – Grant



Agreement nº 291780) and the Junta de Andalucía. We thank Santiago Fernández, Maria Dolores
Hernandez and Antonio Mudarra for their help recovering the core and Inés Morales for the initial core
description and MS data. We thank Jaime Frigola (Universitat de Barcelona) for his help with XRF core
scanning.

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





| Lab Number | Depth (cm) | Dating Method | Age (14C yr BP±1σ) | Calibrated age (cal yr BP)2σ ranges |
|---|---|---|---|---|
| | 0 | Present | 2012 CE | -63 |
| Poz-72421 | 7 | 14C | 40±40 | 29-139 |
| D-AMS 008539 | 22 | 14C | 1112±32 | 935-1078 |
| D-AMS 008540 | 39 | 14C | 2675±30 | 2750-2809 |
| BETA-411994 | 44 | 14C | 3350±30 | 3550-3643 |
| BETA-411995 | 55.5 | 14C | 5480±30 | 6261-6318 |
| Poz-72423 | 57.5 | 14C | 5510±50 | 6266-6405 |
| Poz-72424 | 62 | 14C | 6450±50 | 7272-7433 |
| Poz-72425 | 74 | 14C | 8620±70 | 9479-9778 |

**Table 1.** Age data for LH 12-03. All ages were calibrated using IntCal13 curve (Reimer et al., 2013) with
Clam program (Blaauw, 2010; version 2.2).






















| Correlation | Simulation | | | | | | | |
|---|---|---|---|---|---|---|---|---|
| | A | | B | | C | | D | |
| Ca/Ca (XRF) | 0.63 | p<0.01 | 0.50 | p<0.01 | 0.57 | p<0.01 | 0.54 | p<0.01 |
| K/K (XRF) | 0.53 | p<0.01 | 0.64 | p<0.01 | 0.56 | p<0.01 | 0.65 | p<0.01 |

**Table 2.** Simulation of proxy correlation. A) regular interpolation of 300 years sampling spacing. B)
regular interpolation of 300 years sampling spacing and 5 data points moving average. C) regular
interpolation of 150 years sampling spacing. D) regular interpolation of 150 years sampling spacing and 5
data point moving average.





**984**    **List of figures**

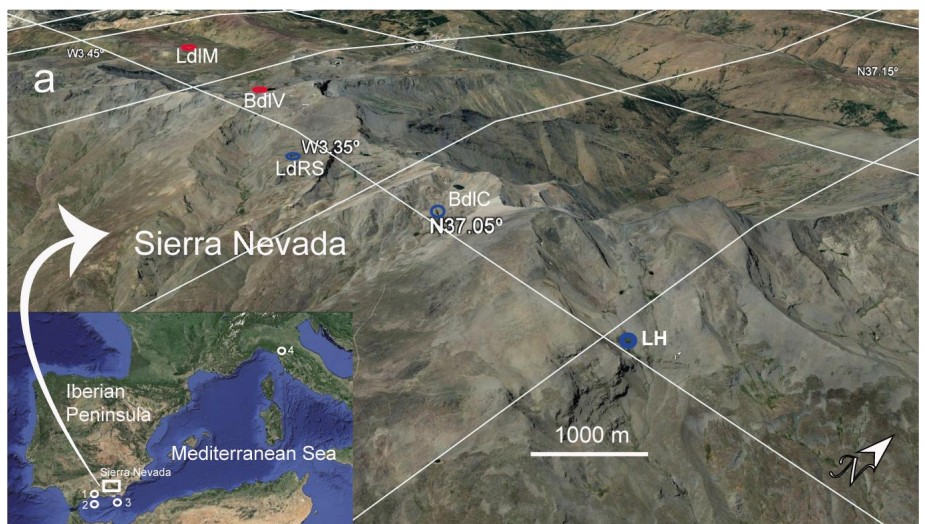

**985**

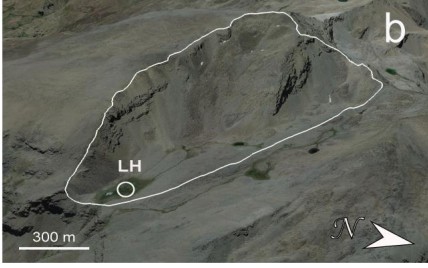
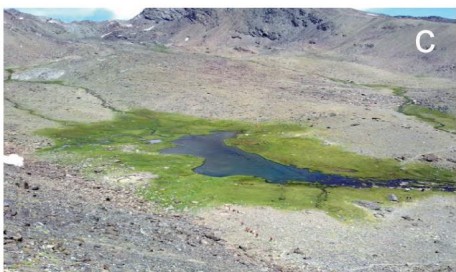

**Figure 1.** (a) Location of the Laguna Hondera (LH) in Sierra Nevada, southern Iberian Peninsula,
Mediterranean region, along with other nearby records   mentioned in the text. (1) El Refugio Cave
stalagmite record; (2) ODP 976 pollen record (Combourieu-Nebout et al., 2009); (3) MD95-2043 pollen
record (Fletcher and Sánchez-Goñi, 2008); (4) CC26, Corchia Cave stalagmite record (Zanchetta et al.,
2007; Regattieri et al., 2014). Sierra Nevada north-facing sites are encircled in red, south-facing sites are
encircled in blue (LH: Laguna Hondera, the current study, is shown in bold). LdLM: Laguna de la Mula
(Jiménez-Moreno et al., 2013); BdLV: Borreguil de la Virgen (García-Alix et al., 2012; Jiménez-Moreno
and Anderson, 2012); LdRS: Laguna de Río Seco (Anderson et al., 2011; García-Alix et al., 2013;
Jiménez-Espejo et al., 2014); BdlC: Borreguil de la Caldera (Ramos-Román et al., 2016; García-Alix et
al., 2017) (b) Regional satellite photo of LH. The catchment area is indicated by the white line. (c) Photo
of Laguna Hondera in September 2012, when the core was taken. Photo taken by Gonzalo Jiménez-
Moreno



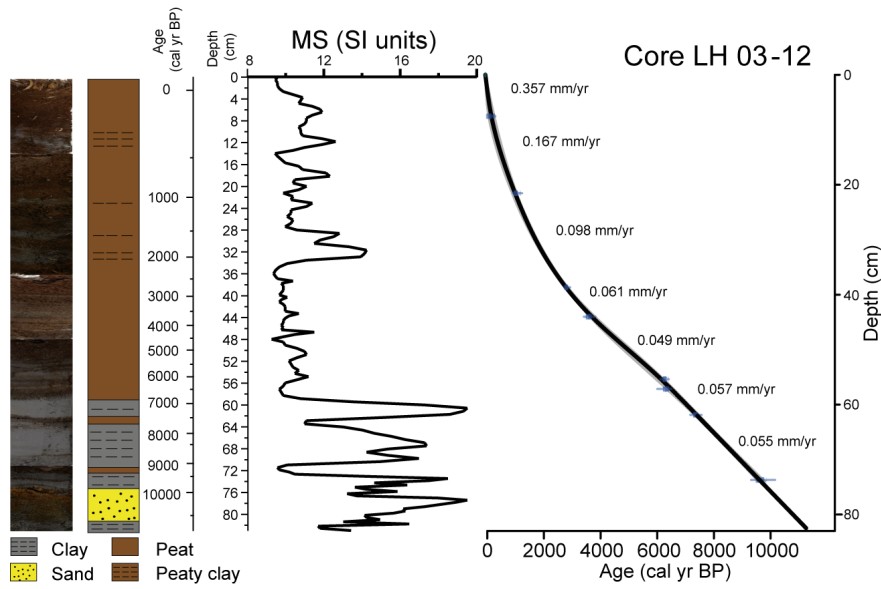


**Figure 2.** Photo of core LH 12-03, along with the lithology, magnetic susceptibility (MS, in SI units)
profile and age-depth model. Sediment accumulation rates (SAR in mm yr $^{-1}$) are shown between
individual radiocarbon ages (see details in text for method of construction).



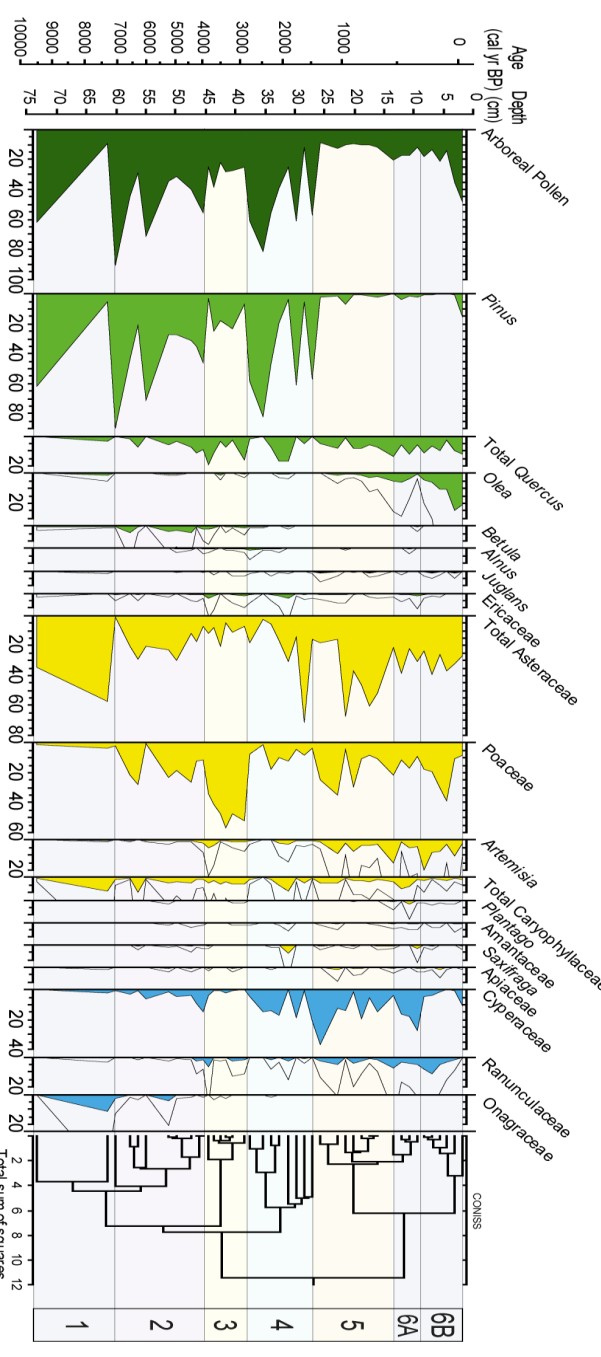

**Figure 3.** Pollen percentage diagram of the LH 12-03 record showing major selected taxa. Major tree species are shown in green; shrubs and herbs are shown in yellow; and wetland and aquatic types are in blue. Pollen was graphed with the Tilia program (Grimm, 1993), and zoned using the CONISS cluster analysis program (Grimm, 1987).



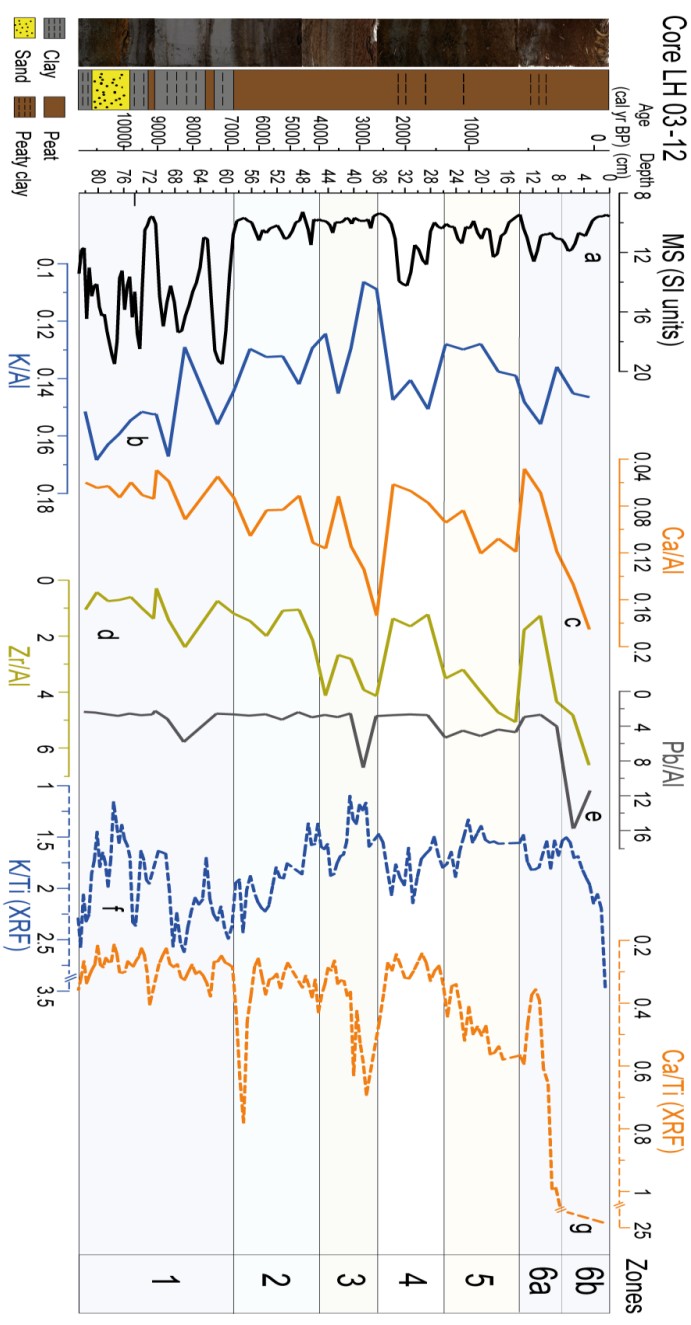


**Figure 4.** Detailed geochemical diagram of the LH 12-03 record showing the selected proxies: (a) MS;
(b) K/Al; (c) Ca/Al; (d) Zr/Al; (e) Pb/Al; (f) K/Al (XRF); (g) Ca/Al (XRF) (MS in SI units, Zr/Al and
Pb/Al scale x $10^{-4}$ and XRF in counts). Pollen zonation described in section 4.3 was used.



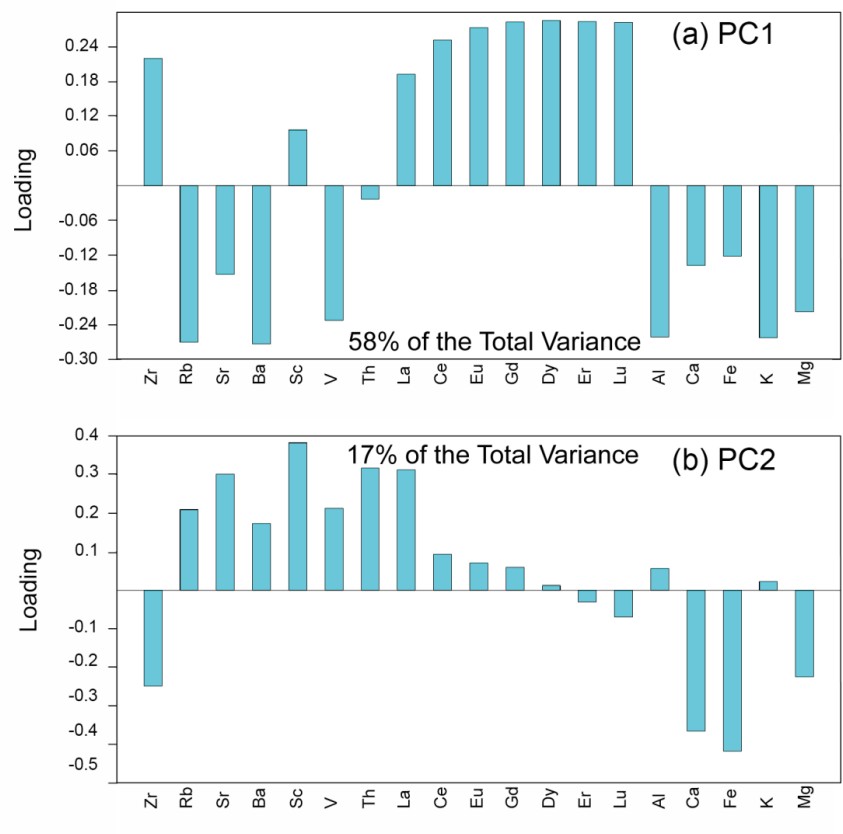

**Figure 5.** Principal Component Analysis (PCA) loadings from selected geochemical elements. (a) PC1,
which describes 58% of total variance; (b) PC2, which describes 17% of total variance.



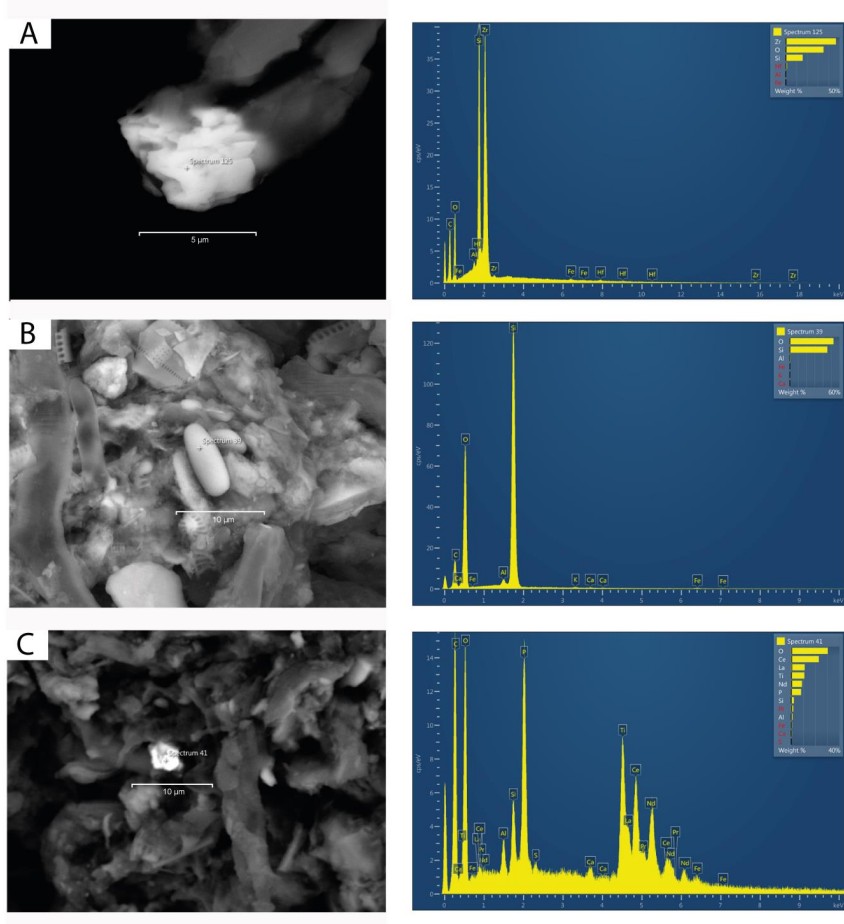


**Figure 6.** Electron Backscatter Diffraction microphotographs of the LH record with clearer colours representing heavier minerals. (a) Zircon, with high Zr content (Dr. 01, 4-5 cm); (b) rounded quartz related with eolian transport (Dr. 01, 2-3 cm); (c) monazite, with high REE content (Dr. 01, 2-3 cm).





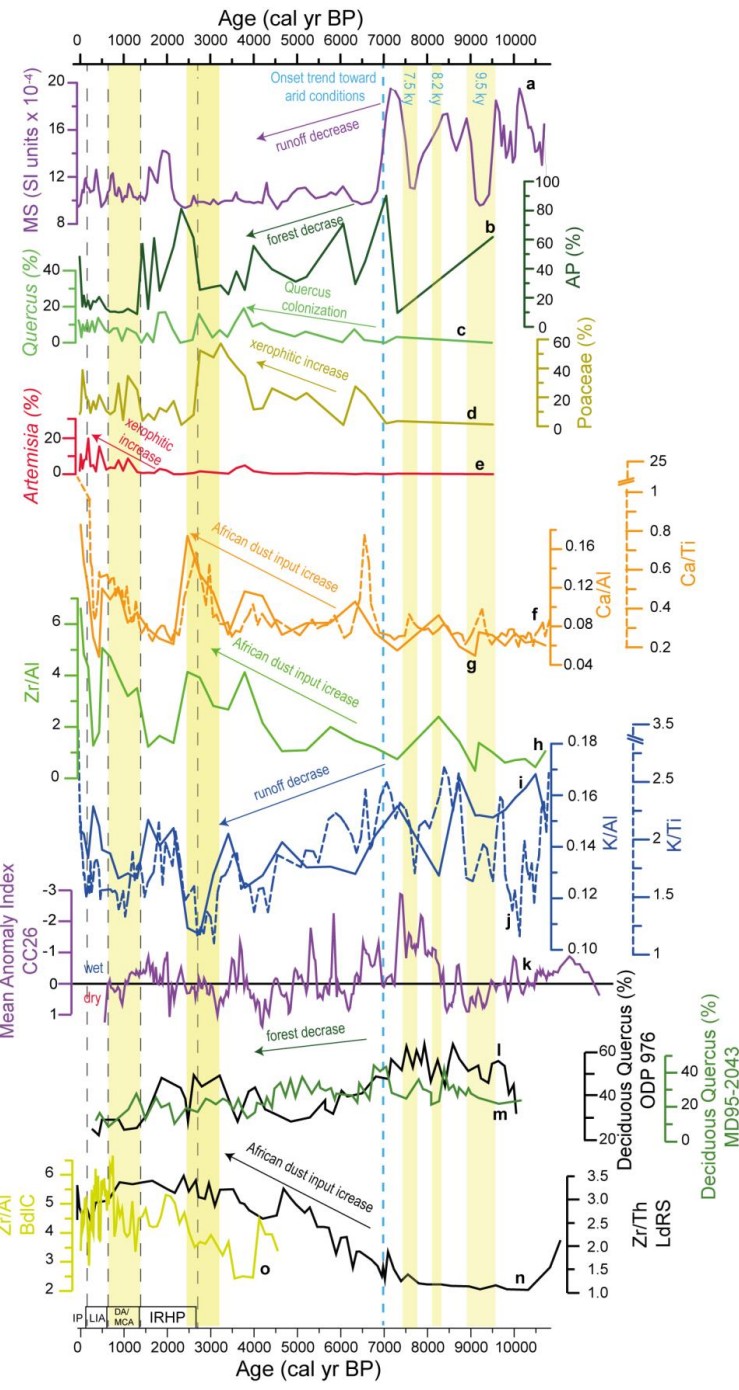

**Figure 7.** Comparison of MS data (in SI units x 10⁻⁴), the most important pollen taxa and geochemical proxies from LH 12-03 record, with nearby paleoclimate records. (a) LH Magnetic Susceptibility (MS) record; (b) Arboreal Pollen (AP) percentage from LH; (c) *Quercus* percentage from LH; (d) Poaceae





percentage from LH; (e) Artemisia percentage from LH; (f) Ca/Ti (XRF) ratio from LH in dashed line;
(g) Ca/Al ratio from LH; (h) Zr/Al ratio from LH; (i) K/Al ratio from LH; (j) K/Ti (XRF) ratio from LH
in dashed line; (k) Mean Anomaly Index from CC26 record (Corchia cave; Regattieri et al., 2014); (l)
Deciduous *Quercus* ODP 976 (Alboran Sea; Combourieu-Nebout et al., 2009); (m) Deciduous *Quercus*
MD95-2043 (Alboran Sea; Fletcher and Sanchez-Goñi, 2008); (n) Zr/Th ratio from Laguna de Río Seco
(LdRS); (o) Zr/Al ratio from Borreguil de la Caldera (BdlC). Yellow bands indicate more arid intervals.
Dark dashed lines are used for separating the different CE periods: IRHP: Iberian Roman Humid Period;
DA: Dark Ages; MCA: Medieval Climate Anomaly; LIA: Little Ice Age; IP: Industrial Period. Blue
dashed line indicates the onset of the trend toward arid conditions.












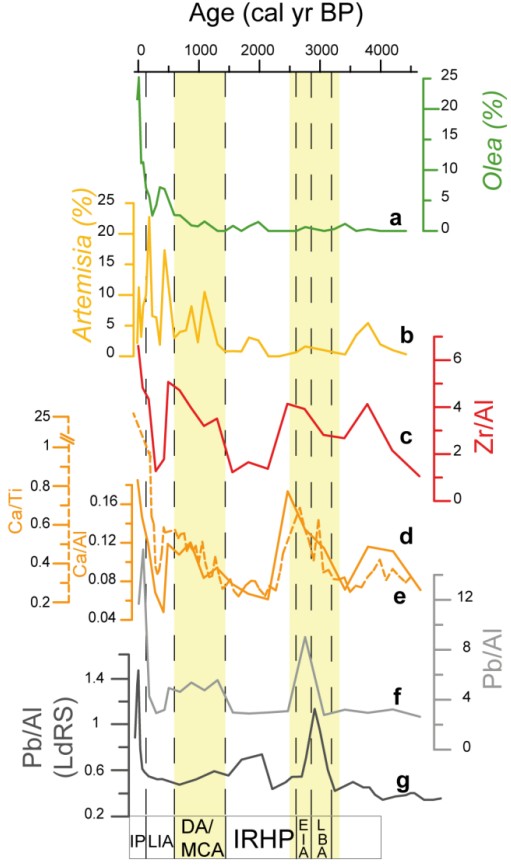


**Figure 8.** Comparison of geochemical proxies with pollen taxa, related to anthropogenic impact for the
last ~4500 cal yr BP. (a) *Olea* percentage from LH; (b) *Artemisia* percentage from LH record; (c) Zr/Al
ratio from LH; (d) Ca/Al ratio from LH; (e) Ca/Ti (XRF) ratio from LH; (f) Pb/Al ratio from LH; (g)
Pb/Al ratio from Laguna de Río Seco (LdRS). Yellow bands indicate more arid intervals. Dark dashed
lines are used for separating the different CE and BCE periods: LBA: Late Broze Age; EIA: Early Iron
Age; IRHP: Iberian Roman Humid Period; DA: Dark Ages; MCA: Medieval Climate Anomaly; LIA:
Little Ice Age; IP: Industrial Period.