# Peer review of "Vegetation and geochemical responses to Holocene rapid 1"

_Climate of the Past, 2018_

## Referee Comment (RC1) · Anonymous Referee #1 · 31 May 2018

Referee Comment for "Vegetation and geochemical responses to Holocene rapid climate change in Sierra Nevada (SE Iberia): The Laguna Hondera record" Author(s): Jose Manuel Mesa-Fernández et al.

1. Does the paper address relevant scientific questions within the scope of CP?

Yes.

2. Does the paper present novel concepts, ideas, tools, or data?

The data is new and confirms to some degree other datasets from southern Spain/the

[Figure]

Mediterranean region.

3. Are substantial conclusions reached?

Yes.

4. Are the scientific methods and assumptions valid and clearly outlined?

Yes, but some methods could be outlined in more detail.

5. Are the results sufficient to support the interpretations and conclusions?

Generally yes.

6. Is the description of experiments and calculations sufficiently complete and precise to allow their reproduction by fellow scientists (traceability of results)?

Yes, but there is room for improvement.

7. Do the authors give proper credit to related work and clearly indicate their own new/original contribution?

Generally yes, but it could be stated even more clearly what is special about the new record in the interpretation and/or how it differentiates from other records.

8. Does the title clearly reflect the contents of the paper?

Yes.

9. Does the abstract provide a concise and complete summary?

Yes.

10. Is the overall presentation well structured and clear?

The paper is well-structured.

11. Is the language fluent and precise?

Generally yes. The English sounded strange to me in some very few cases (mentioned below), but I am not a native speaker.

12. Are mathematical formulae, symbols, abbreviations, and units correctly defined and used?

Yes.

13. Should any parts of the paper (text, formulae, figures, tables) be clarified, reduced, combined, or eliminated?

Yes, in a few cases.

14. Are the number and quality of references appropriate?

Yes.

15. Is the amount and quality of supplementary material appropriate?

The supplementary material could be used to give more information on methods (e.g. pollen differentiation and the used cluster analysis).

Dear authors (and dear editor),

let me first state that I am palynologist, thus, it would be good if a second referee would be a sedimentologist/geochemist with experience concerning the geochemical record, which seems to me to be of more relevance for the manuscript (ms in following) than the pollen record.

I find this manuscript interesting and think that it may be worth publishing. It is already convincing in many aspects, and as I have seen, it has already been improved due to remarks by Nathalie Combourieu Nebout. There are still some general aspects and also some aspects concerning the palynological data which I think should be explained/improved/changed.

One of my main concerns is that bisaccate pollen with its particularly excellent eolian

transport characteristics accounts for 80% of the pollen sum in several samples, and not only in the lower part of the record, which means that the counting sum for the remaining pollen types is only 60 grains... Thus, I wonder how robust the pollen signal is. In a marine record, you would probably exclude Pinus/bisaccate pollen – I am not sure if this would not also make sense in case of Laguna Hondera, since I would expect this pollen type to be overrepresented. At least, you should discuss aspects like eolian transport affecting the pollen data. Eolian transport is often discussed in the MS in context with the geochemical data, but, if I have not missed it, not in context with the pollen signal. I could for example imagine that even minor changes in wind direction may lead to significant signals in the Pinus signals. Maybe there is even the possibility to use the geochemical data to get an idea how wind direction and energy changed over time? It is unfortunate that the pollen preservation and sample density is not high in the lower part of the record, it seems to me that otherwise you could find a correlation between K/Ti and tree pollen. . .

In this context, I also wonder if it is a good move to use the pollen zonation also as a base to describe the geochemical results. Particularly in the lower part, the CONISS-based zonation does not seem to be very relevant considering the resulting dendrogram depicted in Fig. 3. Additionally, I cannot completely follow the zonation as described in the text, I think there are discrepancies with the cluster analysis.

Kind regards.

Below, I make detailed remarks/corrections.

Abstract:

Line 22 "magnetic susceptibility proxies" makes no sense – this sounds as if you measured a proxy for magnetic susceptibility. Rephrase.

Line 22 and following: You state later in the text that the pollen record is not reliable until ∼7000 cal yr BP, yet you state here that palynological proxies indicate humid conditions

until ∼7000 cal yr BP. This should be rephrased.

Generally, I would not introduce abbreviations in the abstract.

2.1. Regional climate and vegetation

Line 79: ". . . which in turn controls. . ." If this relates to "altitudinal contrasts", the verb should be in plural.

Line 86: "(1900 -2800 m)" – blank space behind "-"

Line 86 and Line 89: The "," behind the brackets should be removed.

Line 89: Here, a long dash is used (600 – 1400 m).

2.2. Laguna Hondera

Sometimes there is a blank space between value and unit ("2800 m"), sometimes not ("3366m"): Make it consistent.

3.2 Pollen

Consider my general comment above: The counting sum is generally okay, but with both bisaccate and Poaceae included, it is not that impressive. I must also admit that I find it difficult to separate Olea pollen from, e.g., Phillyrea, at only 400x magnification. Since Olea is important for the discussion and even mentioned in the abstract, and other Oleaceae are not mentioned in the pollen diagram, it might be interesting to show an encountered Olea grain in the supplements and/or to discuss how Olea was differentiated.

Line 127: "Typha" should be in italics.

Line 127/128: ". . . plotted using Tilia program. . ." This does not sound like proper English to me.

Line 129: Maybe a few more sentence to the cluster analyses could help. I was not aware of CONISS until now and learned from Grimm (1987) that normally the used

algorithm operates on a dissimilarity matrix of squared Euclidian distances. Was this the case in your analysis? Did you do an unconstrained analyses to control your results? Generally, I wonder if it makes sense to use the algorithm of Grimm (1987) with a dataset in relatively low resolution and at the same time significant variance between stratigraphically neighboring samples (in the lower part at least), but this is probably a matter of taste and you mention the problem later to some degree...

3.5 Statistical Analyses

Line 156: "PCA finds. . ." Is this proper English?

A few more details may be interesting. What was used, R-mode or Q-mode?

4.2 Chronology and sedimentation rate

Line 169: "The age model" instead of "The age –model".

4.3 Pollen

Generally, I think there are too many pollen zones based on the cluster analyses. Instead of 6-7 zones, I would rather suggest to reduce this to 4 zones considering the distances shown in Fig. 3. Or perhaps you should use another way to define the zones.

Line 183: I am not sure if I misunderstand something here: In Fig. 3, to me there seem to be two samples within Zone LH-1. A third sample is just above the borderline to Zone LH-2. This makes also sense since the dendrogram in Fig. 3 implies that the two lowermost samples are grouped together, while the third sample from below is more similar to the samples in LH-2. One factor which might cause this is the high abundance of Asteraceae in the two lowermost samples. However, in the text, it is mentioned that three samples define the lowermost zone (LH-1), and this is repeated later in line 311. This is of importance since the pollen zonation is also used for the interpretation of the geochemical data. In this context: The zone borderlines in Fig. 4 are at slightly different depths than in Fig. 3 (see below).

Line 186: "takes place" sounds strange to me in this context.

Line 187: "Caryophyllaceae": I am not sure about the grammar here – while you can read many English texts using family names in singular, I still think it should be used as a plural form. In any case, here and in the following cases, it would probably be best to combine the family name with "pollen" or "pollen grains", in this case e.g. "Caryophyllaceae pollen" with singular or "Caryophyllaceae pollen grains" with plural.

Line 191: "Quercus" should be in italics. And (see above) it may be clearer to write "Quercus pollen".

Line 193: "is less" does not sound like proper English to me.

Line 196: "Zone LH-3. . . is defined primarily by a great increase in Poaceae pollen. . ." This is confusing. Is it possible with CONISS to check which parameters influenced the resulting cluster more or less? Or did you make a control test without Poaceae pollen? (You are probably right, but still, I suggest to rephrase this sentence!)

Line 201: You can as well remove "in this zone", it is redundant.

Line 210: Maybe I completely misunderstand how the cluster analysis is used. According to the dendrogram in Fig. 3, Zone LH-6A appears to be more similar to Zone LH-5 than to Zone LH-6B. Following the analyses, LH-6A should rather be LH-5B. . . Or did you have additional reasons to combine 6A and 6B? But then, this should be explained.

Line 210: Here and in Fig. 3, capitalized As and Bs are used for the subszones (LH-6A and LH-6B), but in the results concerning the geochemical data, the subzones are named LH-G6a and LH-G6b. Furthermore, in Fig. 4, there is no "G" in front of the subzone number.

4.4 Sediment composition

Line 218: ". . . makes it important. . ." Sounds strange to me.

Line 224/225: "... or high organic and water content that increase..." Not sure here, but I guess either "contents" or "increases".

Line 249: Remove blank space after "∼". Considering the geochemical data, it may make sense to have a subunit LH-6A, but as mentioned above, I think this is not really supported by the pollen data. Would it make sense to define units using both pollen and geochemical data?

5. Discussion

As written above, I cannot really evaluate the quality of the interpretation of the geochemical record, particularly section 5.3 is not really part of my expertise.

5.1.1.

Line 311: See above: Why three samples?

5.1.2.

Line 383: The mentioned peak in Artemisia is quite weak, I think, and only consists of one sample. There are much stronger peaks above...

5.1.3.

Line 421: I think this is a good example for the "Pinus problem": In the short interval between 2300 and 1800 cal yr BP, I cannot really see the decreasing trend in tree pollen that you postulate, instead, I just see fluctuating tree pollen percentages, and this may be partly caused by pollen transport effects (e.g. changes in wind direction). Maybe the longer interval from 2500 to 1200 cal yr BP in total reflects a decrease in tree pollen (but with fluctuations).

Line 421: "... between 2300-1800...": I would either suggest "between 2300 and 1800 cal yr BP" (add the unit!) or "from 2300 to 1800 cal yr BP". This is also occurring earlier (e.g. Line 384). But maybe it is all right the way used in the MS if it fits with the style of CoP.

5.1.4. and 5.1.5.

I think interpretation becomes "stronger" (if you like) in these sections, not least since the pollen data is probably more robust here. However, concerning section 5.1.4., I wondered if it is not possible to discuss differences between LH and the other mentioned records instead of just mentioning that there were arid conditions "everywhere". What differentiates your pollen record from other records from the western Mediterranean region, and how can these differences be explained?

Conclusions

Line 543: Add "at" before "∼" and remove the blank space behind "∼".

Again, the "between XXXX-XXXX" problem occurs here several times.

Line 554: replace "is" with "are".

Acknowledgements

Line 568: Replace "acknowledge" with "acknowledges".

References

In some cases, there are blank spaces between the initials, in some cases not.

Figures

The abbreviation "LH" should be explained in the figure texts, even if it is already explained in the main text, too, since figure texts should be understandable on their own.

Figure 1:

The figure is quite fancy, but printed (particularly in black and white), it is difficult to see what it is depicted. I would prefer outlined maps instead of the first two photos. But this is rather a remark, since it is a matter of taste.

Figure 2:

I think some texts are too small here, e.g. "Age " and "Depth".

Figure 4:

It is a little confusing that you use 4 cm steps here for the depth scale, but 5 cm in figure 3. The pollen zones seem to be slightly shifted compared to Fig. 3, e.g., the borderline between zone 3 and 4 seems to be at 38 cm in Fig. 3, but 36 cm in Fig. 4. Similarly, the borderline between zones 2 and 1 seems to be at 60 cm in Fig. 3, but at 59 cm in Fig. 4.

The names of the subzones differ from Fig. 3 (see above).

Figure text: "Pollen zonation described in section 4.3 was used." Is this proper English?

Figure 7:

"Quercus" is not in italics in all cases. "Artemisia" is not in italics in the figure text.

Figure 8:

The yellow scales are quite close to each other.

---

## Referee Comment (RC2) · Anonymous Referee #2 · 15 Jul 2018

Dear authors and editor:

I have thoroughly red with much interest the excellent paper of Jose Manuel Mesa-Fernández and his colleagues. The paper deals with climate and environmental reconstruction based on a core record from Laguna Hondera, a small lake located at ∼2900 m above mean sea level in Sierra Nevada (southern Spain). Using a series of parameters and proxies, the authors utilize the sediment record to reconstruct ∼9000 years of both regional climate and environmental variability, while concurrently emphasize the impact of long distance dust transportation from the Sahara into the lake region.

[Figure]

I find the paper extremely interesting and timely for publication. The authors indeed point to highly important conclusions that are relevant to the scope set by the CP journal and of interest for the wide-scale paleoclimate scientific community. Yet, I believe that some points need to be better emphasized or explained in order to strengthen the article and reach the high-impact level as requested by the journal.

Below these lines I portray the main concerns:

1) Chronology and possible presence of hiatuses and/or unconformities in the sedimentary record. The authors do not consider at any step of the article the possibility that the record might not be complete or continuous. Considering the small-scale lake morphology (which unfortunately information on size and depth are missing), I believe that these issues definitely need to be taken into consideration. Furthermore, the authors state that when coring procedures took place, the lake size was even smaller than in previous years, testifying for the great variability such a lake system may have suffered in the past. Regarding the sedimentological record, just by looking at the lithological changes (Figs. 2 and 4) I estimate that depths characterized by sharp lithological variabilities (and that are associated with abrupt petrophysical and geochemical variations; e.g., at ca. 34 cm depth), may also result from interrupted sedimentation. I also think that the age/depth model in figure 2 is misleading as the dots in the graph does not necessary needs to be connected. Moreover, if you decide that the curve should be shown as is, so definitely a plus/minus range (such as a strip in the entire plot) should be given as well.

2) The suggestion that Pb/Al is a reliable recorder for human impact in the region. I sincerely do not see any important or drastic change in this parameter that can be connected with human impact, especially not in the suggested age of ∼2800 cal yr BP (Fig. 8). The authors suggest that anthropogenic impact in the environment can be identified at this time, but this assumption is definitely not sustained by the data. There is only a single peak occurring at this time and not a trend towards increase values, as should be expected. In my opinion, the change on those elements are a direct result of a local variability in the source watershed, which can also be appointed (in this case) to natural causes. Yet, I totally agree with the trend identified for the younger sedimentary sequence (∼150 years BP). The authors indeed explain the possible dilution of Pb in the LH record as a consequence of an increase catchment area and more humid conditions and they rely the assumptions solely on comparison with a nearby record (LdRS). I suggest the authors to abstain of reaching such a concise statement only by comparing with a single site, and neglecting that their own data does not strongly support those evidence.

3) Saharan dust. The idea is definitively well presented and well discussed against datasets from the African continent (although some references are missing, such as Krueger et al. Atmospheric Environment 38, no. 36 (2004): 6253-6261). Yet, I argue that the increase in Ca in their record (Fig. 7) does not necessary need to only imply dust coming from the Sahara. I suggest the authors to also consider the possibility that during increase intervals of Ca/Al or Ca/Ti (especially ∼3300-2500 cal yrs ago, according to their figure), Ca elements could derive from exposed areas of continental shelves in southern Spain or Northern Africa. Those regions will probably include much greater amount of Ca, when compared with datasets from the Sahara. Please refer to the necessary literature for discussing this issue.

I have made further corrections directly on the PDF associated with this letter.

In light of the major comments listed above and those directly written on the PDF, I recommend the authors to carry out a comprehensive re-structuralization of their paper prior to further acceptance by the editor.

Kind regards,

Reviewer #2

Please also note the supplement to this comment:
https://www.clim-past-discuss.net/cp-2018-35/cp-2018-35-RC2-supplement.pdf

[Figure]

**Supplement:**

[revised manuscript text omitted]

---

## Author Comment (AC1) · 9 Aug 2018

Author's response to Referee #1

We thank Referee # 1 for such a detailed and careful review and for all constructive comments and suggestions. These have been taken all into consideration and have significantly contributed to improve the quality of the revised version. We list below comments and corresponding responses to each point raised.

The Referee's comment below are in italics and our answer in bold font.

- *One of my main concerns is that bisaccate pollen with its particularly excellent eolian transport characteristics accounts for 80% of the pollen sum in several samples, and not only in the lower part of the record, which means that the counting sum for the remaining pollen types is only 60 grains... Thus, I wonder how robust the pollen signal is. In a marine record, you would probably exclude Pinus/bisaccate pollen – I am not sure if this would not also make sense in case of Laguna Hondera, since I would expect this pollen type to be overrepresented. At least, you should discuss aspects like eolian transport affecting the pollen data.*
  **We agree that bisaccate pollen is indeed favoured by wind transport and has a larger dispersal area than other tree species. Nevertheless, the higher concentrations of *Pinus* in the LH record make sense as they occurred during the Early Holocene and the Iberian Roman Humid Period (IRHP), the two warmest and most humid periods in the whole Holocene. Laguna Hondera is located at 2899 masl only 99 m above of the upper boundary of the oromediterranean belt (between around 1900-2800 masl) where *Pinus sylvestris* is the main tree species. Therefore, this apparently anomalous high concentration may be caused by an upward migration of the oromediterranean belt and treeline towards higher elevations and around the LH during these warmer periods, which could have been overstate due to the high pollen-production and dispersal of *Pinus*. For this reason we think that we should not exclude *Pinus*/bissacate pollen from the total sum, because we are recording a regional climatic signal, without allocthonous influence. In any case, we have expanded the discussion about the possible effect of enhanced wind conditions during those times, affecting the amount of *Pinus* pollen input into the LH site (see comment below).**

- *Eolian transport is often discussed in the MS in context with the geochemical data, but, if I have not missed it, not in context with the pollen signal. I could for example imagine that even minor changes in wind direction may lead to significant signals in the Pinus signals. Maybe there is even the possibility to use the geochemical data to get an idea how wind direction and energy changed over time? It is unfortunate that the pollen preservation and sample density is not high in the lower part of the record, it seems to me that otherwise you could find a correlation between K/Ti and tree pollen.*
  **This comment about wind direction changes is very interesting. However, it is difficult to find proxies to solve this question. The oromediterranean vegetation belt, rich in *P. sylvestris*, occurs in Sierra Nevada in all directions surrounding the studied site and if any change in wind direction occurred, the lake would have received the same amount of *Pinus* input (See Fig.1). Furthermore, there is a moderate anticorrelation between *Pinus* and Zr (r=-0.51; p=<0.01) and no correlation between *Pinus* and Ca (r=-0.12; p=0.46). This suggests that the abundance of *Pinus* pollen in our lake sediments is not linked to the enhancement of southern winds (which are related with arid conditions) but to warm and humid climatic condition favouring *Pinus* development in the surrounding area. Nevertheless, one factor that could have changed the amount of *Pinus* input in our**

**record is changes in the wind energy. Since persistent negative NAO results in more humid conditions and higher westerlies influence over southern Europe, the higher presence of *Pinus* in the surrounding area, along with the higher wind energy over Sierra Nevada could have resulted in more bissacate pollen input into our lake. This is consistent with the anomalous high percentages of *Pinus* recorded during the IRHP and support the significance of our pollen data. There is no correlation between *Pinus* and K distribution or the K/Ti ratio, likely because of the reasons that you exposed in your question. We have included a new paragraph in the manuscript with some more discussion about this, lines 268-276.**

- *In this context, I also wonder if it is a good move to use the pollen zonation also as a base to describe the geochemical results. Particularly in the lower part, the CONISS based zonation does not seem to be very relevant considering the resulting dendrogram depicted in Fig. 3.*

  **Yes, we fully agree. We rewrote the geochemical results without using the pollen zonation. See section 4.4.**

  *Additionally, I cannot completely follow the zonation as described in the text, I think there are discrepancies with the cluster analysis.*

  **We have simplified the zonation in order to make the text more comprehensible. See discussion below.**

*Others remarks about the text.*

- *Line 22 "magnetic susceptibility proxies" makes no sense – this sounds as if you measured a proxy for magnetic susceptibility. Rephrase.* **We rephrased this sentence, see lines 22-23.**
- *Line 22 and following: You state later in the text that the pollen record is not reliable until ~7000 cal yr BP, yet you state here that palynological proxies indicate humid conditions until ~7000 cal yr BP. This should be rephrased. Generally, I would not introduce abbreviations in the abstract.* **We totally agree, we removed "palynological proxies" and rephrased the sentence. Abbreviations were removed from the abstract, see lines 22-24.**

2.1. Regional climate and vegetation

- *Line 79: "… which in turn controls…" If this relates to "altitudinal contrasts", the verb should be in plural.* **We rewrote the verb in plural. See line 78.**
- *Line 86: "(1900 -2800 m)" – blank space behind "-"* **The blank space was removed, see line 84.**
- *Line 86 and Line 89: The "," behind the brackets should be removed.* **The "," was removed. See lines 85 and 87.**
- *Line 89: Here, a long dash is used (600 – 1400 m).* **We changed it to short dash, see line 87.**

2.2. Laguna Hondera

- *Sometimes there is a blank space between value and unit ("2800 m"), sometimes not ("3366m"): Make it consistent.* **We put a blank space between value and unit. See lines 96-102.**

3.2 Pollen

- *Consider my general comment above: The counting sum is generally okay, but with both bisaccate and Poaceae included, it is not that impressive. I must also admit that I find it difficult to separate Olea pollen from, e.g., Phillyrea, at only 400x magnification. Since Olea is important for the discussion and even mentioned in the abstract, and other Oleaceae are not mentioned in the pollen diagram, it might be interesting to show an encountered Olea grain in the supplements and/or to discuss how Olea was differentiated.* **We agree that sometimes it is difficult to differentiate *Olea* from other Oleaceae at 400x magnification. We added a paragraph explaining how we differentiated *Olea* and also added a reference. See lines 127-129.**
- Line 127: "Typha" should be in italics. **"Typha" was changed by "*Typha*". See line 124.**
- *Line 127/128: "… plotted using Tilia program…" This does not sound like proper English to me.* **We rewrote the sentence. See lines 125.**
- *Line 129: Maybe a few more sentence to the cluster analyses could help. I was not aware of CONISS until now and learned from Grimm (1987) that normally the used algorithm operates on a dissimilarity matrix of squared Euclidian distances. Was this the case in your analysis? Did you do an unconstrained analyses to control your results? Generally, I wonder if it makes sense to use the algorithm of Grimm (1987) with a dataset in relatively low resolution and at the same time significant variance between stratigraphically neighboring samples (in the lower part at least), but this is probably a matter of taste and you mention the problem later to some degree...* **We specified that we used an age constrained analysis. See lines 126.**
- *Was this the case in your analysis?* **Yes, the program generates a dissimilarity matrix of squared Euclidean distance between the samples.**
- *Did you do an unconstrained analyses to control your results? Generally, I wonder if it makes sense to use the algorithm of Grimm (1987) with a dataset in relatively low resolution and at the same time significant variance between stratigraphically neighboring samples (in the lower part at least), but this is probably a matter of taste and you mention the problem later to some degree...* **We did an age constrained analysis. Visual observations confirm CONISS analyses, so we believe that CONISS works for this dataset of samples. We used CONISS because it seems to work well in most of the published pollen studies, which reconstruct fossil pollen assemblages using it.**

3.5 Statistical Analyses
- *Line 156: "PCA finds…" Is this proper English? A few more details may be interesting. What was used, R-mode or Q-mode?* **We rewrote this sentence and specified the mode used. See lines 153-154. The mode used was R-mode because we applied the PCA to variables, not to samples.**

4.2 Chronology and sedimentation rate
- *Line 169: "The age model" instead of "The age –model".* **We removed the "–". See line 167.**

4.3 Pollen
- *Generally, I think there are too many pollen zones based on the cluster analyses. Instead of 6-7 zones, I would rather suggest to reduce this to 4 zones considering the distances shown in Fig. 3. Or perhaps you should use another way to define the zones.* **We agree with this suggestion and we reduced to 4 the number of pollen zones.**
- *Line 183: I am not sure if I misunderstand something here: In Fig. 3, to me there seem to be two samples within Zone LH-1. A third sample is just above the borderline to Zone LH-2. This makes also sense since the dendrogram in Fig. 3 implies that the two lowermost samples are grouped together, while the third sample from below is more similar to the samples in LH-2. One factor which might cause this is the high abundance of Asteraceae in the two lowermost samples. However, in the text, it is mentioned that three samples define the lowermost zone (LH-1), and this is repeated*

*later in line 311. This is of importance since the pollen zonation is also used for the interpretation of the geochemical data. In this context: The zone borderlines in Fig. 4 are at slightly different depths than in Fig. 3 (see below).* **We changed the zonation (see above) and we decided not to use the pollen zonation for defining the geochemical results.**

- *Line 186: "takes place" sounds strange to me in this context.* **We changed it by "is identified". See line 186.**
- *Line 187: "Caryophyllaceae": I am not sure about the grammar here – while you can read many English texts using family names in singular, I still think it should be used as a plural form. In any case, here and in the following cases, it would probably be best to combine the family name with "pollen" or "pollen grains", in this case e.g. "Caryophyllaceae pollen" with singular or "Caryophyllaceae pollen grains" with plural.* **We agree, we used the plural form in the entire manuscript, because we are alluding to "(Caryophyllaceae) pollen grains".**
- *Line 191: "Quercus" should be in italics. And (see above) it may be clearer to write "Quercus pollen".* **We wrote *Quercus* in italics. See line 189.**
- *Line 193: "is less" does not sound like proper English to me.* **We rewrote this sentence. See line 190.**
- *Line 196: "Zone LH-3:… is defined primarily by a great increase in Poaceae pollen: …" This is confusing. Is it possible with CONISS to check which parameters influenced the resulting cluster more or less? Or did you make a control test without Poaceae pollen? (You are probably right, but still, I suggest to rephrase this sentence!)* **We rephrased it.**
- *Line 201: You can as well remove "in this zone", it is redundant.* **We removed it.**
- *Line 210: Maybe I completely misunderstand how the cluster analysis is used. According to the dendrogram in Fig. 3, Zone LH-6A appears to be more similar to Zone LH-5 than to Zone LH-6B. Following the analyses, LH-6A should rather be LH-5B… Or did you have additional reasons to combine 6A and 6B? But then, this should be explained.* **We changed the zonation according to your previous suggestion.**
- *Line 210: Here and in Fig. 3, capitalized As and Bs are used for the subszones (LH-6A and LH-6B), but in the results concerning the geochemical data, the subzones are named LH-G6a and LH-G6b. Furthermore, in Fig. 4, there is no "G" in front of the subzone number.* **We removed the zonation concerning the geochemical data (see the previous comment).**

4.4 Sediment composition

- *Line 218: "… makes it important…" Sounds strange to me.* **We rephrased it. See line 213.**
- *Line 224/225: "… or high organic and water content that increase…" Not sure here, but I guess either "contents" or "increases".* **We corrected the orthography. See line 219.**
- *Line 249: Remove blank space after "~". Considering the geochemical data, it may make sense to have a subunit LH-6A, but as mentioned above, I think this is not really supported by the pollen data. Would it make sense to define units using both pollen and geochemical data?* **We decided do not to use the pollen zonation for defining the geochemical data.**

5. Discussion

5.1.1.

- *Line 311: See above: Why three samples?* **In this case we are describing the interval between 10800 and 7000 cal yr BP and this period only includes three pollen samples.**

5.1.2.

- *Line 383: The mentioned peak in Artemisia is quite weak, I think, and only consists of one sample. There are much stronger peaks above…* **We agree and removed the mention to this *Artemisia* peak as suggested.**

5.1.3.

- *Line 421: I think this is a good example for the "Pinus problem": In the short interval between 2300 and 1800 cal yr BP, I cannot really see the decreasing trend in tree pollen that you postulate, instead, I just see fluctuating tree pollen percentages, and this may be partly caused by pollen transport effects (e.g. changes in wind direction). Maybe the longer interval from 2500 to 1200 cal yr BP in total reflects a decrease in tree pollen (but with fluctuations).* **We rewrote this sentence, but we have no enough information supporting the "changes in the wind direction" hypothesis. See line 432-438.**
- *Line 421: "... between 2300-1800..." I would either suggest "between 2300 and 1800 cal yr BP" (add the unit!) or "from 2300 to 1800 cal yr BP". This is also occurring earlier (e.g. Line 384). But maybe it is all right the way used in the MS if it fits with the style of CoP.* **We changed the dash between ages by "and" thoughout the entire manuscript.**

5.1.4. and 5.1.5.

- *I think interpretation becomes "stronger" (if you like) in these sections, not least since the pollen data is probably more robust here. However, concerning section 5.1.4., I wondered if it is not possible to discuss differences between LH and the other mentioned records instead of just mentioning that there were arid conditions "everywhere". What differentiates your pollen record from other records from the western Mediterranean region, and how can these differences be explained?* **We have compared our record with other Iberian records showing opposite trends and we have justified these discrepancies. We have also included new references. See lines 462-470.**

Conclusions

- *Line 543: Add "at" before "~" and remove the blank space behind "~". Again, the "between XXXX-XXXX" problem occurs here several times.* **We add "at" before "~" and removed the blank space behind "~". See line 538.**
- *Line 554: replace "is" with "are".* **We removed this sentence.**

Acknowledgements

- *Line 568: Replace "acknowledge" with "acknowledges".* **We replaced "acknowledge" with "acknowledges". See line 562.**

References

- *In some cases, there are blank spaces between the initials, in some cases not.* **We made it consistent.**

Figures

- *The abbreviation "LH" should be explained in the figure texts, even if it is already explained in the main text, too, since figure texts should be understandable on their own.* **We have explained the abbreviations.**

Figure 1:

- *The figure is quite fancy, but printed (particularly in black and white), it is difficult to see what it is depicted. I would prefer outlined maps instead of the first two photos. But this is rather a remark, since it is a matter of taste.* **We appreciate the comment but in online versions figures are in color. In any case we have added a sentence referring the reader to the web version. See lines 1032-1033.**

Figure 2:

- *I think some texts are too small here, e.g. "Age " and "Depth".* **We have enlarged them.**

Figure 4:

- *It is a little confusing that you use 4 cm steps here for the depth scale, but 5 cm in figure 3. The pollen zones seem to be slightly shifted compared to Fig. 3, e.g., the borderline between zone 3 and 4 seems to be at 38 cm in Fig. 3, but 36 cm in Fig. 4. Similarly, the borderline between zones 2 and 1 seems to be at 60 cm in Fig. 3, but at 59 cm in Fig. 4. The names of the subzones differ from Fig. 3 (see above). Figure text: "Pollen zonation*

*described in section 4.3 was used." Is this proper English?* **We used 5 cm steps for the depth scale of both, figures 3 and 4. The zonation was removed in order to simplify the figure.**

Figure 7:

- *"Quercus" is not in italics in all cases. "Artemisia" is not in italics in the figure text.* **We changed them to italics**

Figure 8:

- *The yellow scales are quite close to each other.* **We separated the yellow scales.**

---

## Author Comment (AC2) · 9 Aug 2018

Author's response to Referee #2

We thank Referee #2 his comprehensive comments and suggestion that have allowed us to improve the manuscript. Below we respond to all the general comments, and we accepted all the corrections made in the attached PDF.

The Referee's comment below are in italics and our answer in bold font.

- *(1) Chronology and possible presence of hiatuses and/or unconformities in the sedimentary record. The authors do not consider at any step of the article the possibility that the record might not be complete or continuous. Considering the small-scale lake morphology (which unfortunately information on size and depth are missing), I believe that these issues definitively need to be taken into consideration. Furthermore, the authors state that when coring procedures took place, the lake size was even smaller than in previous years, testifying for the great variability such a lake system may have suffered in the past. Regarding the sedimentological record, just by looking at the lithological changes (Figs. 2 and 4) I estimate that depths characterized by sharp lithological variabilities (and that are associated with abrupt petrophysical and geochemical variations; e.g., at ca. 34 cm depth), may also result from interrupted sedimentation. I also think that the age/depth model in figure 2 is misleading as the dots in the graph does not necessary needs to be connected. Moreover, if you decide that the curve should be shown as is, so definitively a plus/minus range (such as a strip in the entire plot) should be given as well.*

  **Although sharp lithological, magnetic susceptibility (MS) and geochemical changes are recognized throughout the sediment core, i.e. at 58 cm and 34 cm, we believe there is no significant hiatuses or unconformities in our record. For example, at 58 cm the lithology changed from mostly clay to peat, coinciding with the onset of an aridity trend (see section 5.1.2). Nevertheless, the calculated sedimentation rates, deduced from the radiocarbon dated samples, did not show any significant changes, for this reason we consider that sedimentation was fairly continuous. In contrast, at 34 cm the lithology changes from peat to more detritic sediments as a result of a change to more humid climate (the Iberian Roman Humid Period, see section 5.1.3). In this case, the lithology supports more erosion in the basin, enhanced sedimentary input into the lake and likely higher water level. Although a significant increase in the sedimentation rate is recorded from 44 cm to the top, we think that it is most likely due to the lower compaction of the sediments at the top of the record. Thus we consider that no hiatuses occurred during this interval either.**
  **With respect to the question about the age model, the "Clam" software output gives a plus/minus range, represented in figure 2 by the gray shadow. We agree that it was difficult to see and we have made the black line thinner, which shows the age model error more clearly and we have added it to the legend in figure 2.**

- *2) The suggestion that Pb/Al is a reliable recorder for human impact in the region. I sincerely do not see any important or drastic change in this parameter that can be connected with human impact, especially not in the suggested age of _2800 cal yr BP (Fig. 8). The authors suggest that anthropogenic impact in the environment can be identified at this time, but this assumption is definitively not sustained by the data. There is only a single peak occurring at this time and not a trend towards increase values, as should be expected. In my opinion, the change on those elements are a direct result of a local variability in the source watershed, which can also be appointed (in this case) to natural causes. Yet, I totally agree with the trend identified for the younger sedimentary sequence (_150 years BP). The authors indeed explain the possible dilution of Pb in the LH record as a consequence of an increase catchment area and*

*more humid conditions and they rely the assumptions solely on comparison with a nearby record (LdRS). I suggest the authors to abstain of reaching such a concise statement only by comparing with a single site, and neglecting that their own data does not strongly support those evidence.* **We agree with the reviewer that our data is not strong enough for supporting that hypothesis. Instead of covering the last two millennia, we focused our discussion in the last 150 year BP and renamed the section. See lines 489-500. Also, we modified the Figure 8.**

- *3) Saharan dust. The idea is definitively well presented and well discussed against datasets from the African continent (although some references are missing, such as Krueger et al. Atmospheric Environment 38, no. 36 (2004): 6253-6261). Yet, I argue that the increase in Ca in their record (Fig. 7) does not necessary need to only imply dust coming from the Sahara. I suggest the authors to also consider the possibility that during increase intervals of Ca/Al or Ca/Ti (especially _3300-2500 cal yrs ago, according to their figure), Ca elements could derive from exposed areas of continental shelves in southern Spain or Northern Africa. Those regions will probably include much greater amount of Ca, when compared with datasets from the Sahara. Please refer to the necessary literature for discussing this issue. I have made further corrections directly on the PDF associated with this letter. In light of the major comments listed above and those directly written on the PDF, I recommend the authors to carry out a comprehensive re-structuralization of their paper prior to further acceptance by the editor.*

- **We agree with the reviewer that we cannot fully confirm that Saharan dust is the exclusive source of Ca in the LH sedimentary record. In fact, the correlation between Ca and Zr (r =0.57; p<0.05) suggest that Ca likely came from other sources apart from Saharan dust. Previous studies show that the Saharan dust represents the 85% of total dust input in three lowland sites between 500 and 1000 masl around Sierra Nevada (Morales-Baquero and Pérez-Martínez, 2016). Therefore, higher Saharan dust input is expected at higher altitudes since its transport mostly occur between 1500 and 4000 masl (Pulido-Villena et al., 2006; Jiménez et al., 2018). According to Pulido-Villena et al. (2006), the present-day Ca input into the nearby Laguna de Río Seco site from Saharan dust (input to the catchment area plus direct input to lake surface area) has been estimated around 190 kg Ca per year, an amount sufficient to account for the entire lake Ca budget. In addition, the Saharan dust bears others nutrients such as P and N that contribute to microorganism bloom in the Sierra Nevada oligotrophic lakes (e. g. Morales-Baquero et al., 2006, 2013). Jiménez et al. (2018) showed a relationship of the Ca content in the Saharan dust recorded in six Sierra Nevada's lakes, with the increase in *Daphnia*, which use Ca for developing its exoskeleton. Hence we consider that most of the Ca input to LH derives from Saharan dust, but we added a new paragraph in the text highlighting that other different sources could have also been involved. On the other hand, we thoroughly read *Kreuger et al., Atmospheric Environment 38, no. 36 (2004): 6253-6261*, and although it is a great summary of the different eolian dust sources around the world, we instead decided to include *Moreno et al., 2006, Chemosphere, 65, 261-270*, which summarized different African dust sources and its compositions, since they are closer to our record. See lines 288-300.**

**References:**

- Jiménez, L., Rühland, K. M., Jeziorski, A., Smol, J. P., and Pérez-Martínez, C.: Climate change and Saharan dust drive recent cladoceran and primary production changes in remote alpine lakes of Sierra Nevada, Spain, Glob. Change Biol., 24, e139-e158, DOI:10.1111/gcb.13878, 2018.

- Morales-Baquero R., Pulido-Villena, E., Romera, O., Ortega-Retuerta, E., Conde-Porcuna, J. M., Pérez-Martinez, C., and Reche, I.: Significance of atmospheric deposition to freswater ecosystems in the southern Iberian Peninsula, Limnetica, 25, 171-180, 2006.

- Morales-Baquero, R., Pulido-Villena, E., and Reche, I.: Chemical signature of Saharan dust on dry and wet atmospheric deposition in the south-western Mediterranean region, Tellus B, 65, DOI:10.3402/tellusb.v65i0.18720, 2013.

- Morales-Baquero, R., and Pérez-Martínez C.: Saharan versus local influence on atmospheric aerosol deposition in the southern Iberian Peninsula: Significance for N and P inputs, Global Biogeochem. Cycles, 30, 501-513,DOI:10.1002/2015GB005254, 2016.

- Moreno, T., Querol, X., Castillo, S., Alastuey, A., Cuevas, E., Herrmann, L., Mounkaila, M., Elvira, J., Gibbons, W.: Geochemical variations in aeolian mineral particles from the Sahara–Sahel Dust Corridor. Chemosphere, 65, 261-270, DOI:10.1016/j.chemosphere.2006.02.052, 2006.

---

## Author Response (AR1)

Dr. Nathalie Combourieu-Nebout, Editor Climate of the Past (Ref. cp-2018-35)

Dear Nathalie,

Thank so much for considering the resubmission of our manuscript entitled *"Vegetation and geochemical responses to Holocene rapid climate change in Sierra Nevada (SE Iberia): The Laguna Hondera record"* authored by J. M. Mesa-Fernández, G. Jiménez-Moreno, M. Rodrigo-Gámiz, A. García-Alix, F. J. Jiménez-Espejo, R. Scott Anderson, F. Martínez-Ruiz, J. Camuera and M. J. Ramos-Román.
We have gone in detail over all the very constructive suggestions that the two reviewers made, which improved our manuscript to a much stronger paper. The amendments are presented in track changes mode.

Yours sincerely,

Jose Manuel Mesa Fernández

[revised manuscript text omitted]

---

## Author Response (AR2)

21 September 2018

**Dr. Nathalie Combourieu-Nebout**

**Editor Climate of the Past**

Dear Nathalie,

Thank very much for your constructive comment, all of them have been take into account. We hope that the manuscript now fulfil the journal requirements. Here you have a point by point answer (in **bold**) to your comments (in *italics*). We attached the track mode version as supplementary.

- *First, in all diagram is it possible to report the scale (cm and or cal yr) on the two sides (left and right or top and bottom), it will be easier to see correlations between the different curves. This has to be done on fig 3, 4, 8.* **We included the scale on the two sides of each figure.**

- *Figure 2: it remains very difficult to see the grey shadow. Perhaps with another color it will be more readable.* **We changed it to red to make it more readable.**

- *Figure 3: The problem of pine has been underline and you respond satisfactorily to the reviewer but I do not see the additional paragraph at the place indicated in your response. Probably it correspond to the paragraph between line 291 and 299. I understand that you interpret the high percentage of Pine as a migration of the tree line. But for me it is not sufficient for the third sample. In fact, I have a doubt on the third sample as this sample is almost completely full of pine. Do you really think it is reliable? In fact it is very strange to have almost only Pine in a sample of a continental site. Does this sample not rather reflect a problem of differential grain conservation? If it is the case, this sample has to be removed from the diagram or at least it has to be more extensively discussed if you keep this sample because it remains highly questionable. As the peak is defined and by one sample composed only by pine it has to be taken with cautious in my opinion. The following peaks do not show the same picture and seems more reliable even if defined by one sample.* **First of all, I am very sorry that the paragraph did not coincide with the lines indicated in the author's response file, the manuscript changed its format when I opened it with other PC and I did not noticed it. On the other hand, the third sample definitely present a preferential conservation of *Pinus*, so we decided to remove it. We also made the corresponding changes in the manuscript. See lines 186; 188; 191.**

- *Figure 6: I do not understand what are the small dendrograms in the upper right corner of each blue figure. In fact they are too little to be red. Remove them if it is not used in the paper or enlarge them to propose them in a readable way.* **The dendrograms indicate the elemental composition of each mineral, we enlarged them in order to make it more readable. See line 1052.**

- *Figure 7: Decrease is not written in a good way (decrase) for forest, African dust and same for increase (icrease). Please verify all. The characters in yellow, light green and light orange are very difficult to read but perhaps it is a printing problem. Verify that please. When you mark runoff decrease it correspond to an abrupt stop of the runoff at*

*the beginning of the arrow. Wouldn't it be better to write stop and not decrease?* **We corrected the misspellings and we replaced the light colors for darker ones. Since there is no a runoff stop, we changed "runoff decrease" for "more peaty lithology" which also implies a runoff decrease.**

- *In the text, you added something on Olea but it is not clear for me. For you which pollen have the thicker endexine (and not intine which is not present in the past pollen) and which one has the higher size of reticulum? I think it is Olea and not Phyllirea. Please rephrase you sentence. In the same paragraph, I think that precision are needed on the way you calculate the percentages. What does the basic pollen sum contain: all the grains or only the 300 without aquatic? Is pine included in it (probably)? You have to precise that. Please add something.* **We rephrased both paragraph. See lines 126-129 and 131-132**

Kind regards,

Jose